# Trifluoroacetate (TFA) in precipitation and surface waters in Switzerland: trends, source attribution, and budget TS1

**Stephan Henne[1], Florian R. Storck[2], Henry Wöhrnschimmel[2], Markus Leuenberger[3], Martin K. Vollmer[1], and Stefan Reimann[1]**

[1]Empa, Laboratory for Air Pollution/Environmental Technology, Dübendorf, Switzerland
[2]FOEN, Federal Office for the Environment, Bern, Switzerland
[3]Climate and Environmental Physics Division and TS2 Oeschger Centre for Climate Change Research, University of Bern, Bern, Switzerland

**Correspondence:** Stephan Henne (stephan.henne@empa.ch)

**Abstract.** Sources and budgets of the persistent, anthropogenic compound trifluoroacetate (TFA) are poorly quantified across different environmental media. Recently, the introduction of hydrofluoroolefins and the continued use of other fluorinated compounds has increased environmental levels of TFA. Here, we present concentrations of TFA observed in precipitation and surface waters in Switzerland during three years of continuous monitoring and in archived water samples, collected since 1984. Mean observed TFA concentrations ranged from 0.30 to 0.96 $\mu g\,L^{-1}$ across 14 precipitation sites and from 0.33 to 0.88 $\mu g\,L^{-1}$ across 9 river sites in 2021–2023 – a four-to-six-fold increase since 1996/1997. Simulated atmospheric degradation of known TFA precursors accounted for 63 % (58 %–70 %) of the observed deposition (48 % (41 %–54 %) hydrofluoroolefins and 15 % (12 %–18 %) long-lived fluorinated gases; mean and range across sites) for sites on the Swiss Plateau. In Switzerland, atmospheric (wet+dry) deposition of TFA amounted to $24.5 \pm 9.6\,Mg\,yr^{-1}$, whereas TFA terrestrial inputs from the degradation of plant protection products (PPP) and veterinary pharmaceuticals in soils, estimated from the literature, ranged from 3.9 to $13.2\,Mg\,yr^{-1}$, depending on the assumption on degradation efficiency. TFA inputs from the degradation of PPP dominated 2–3 times over atmospheric deposition in Swiss croplands. These inputs were balanced by exports through the major rivers, $31 \pm 4\,Mg\,yr^{-1}$. Archived precipitation samples from the period 1986 to 2020 revealed that TFA was formed in the atmosphere before the introduction of known atmospheric precursors, whereas in the 1990s TFA deposition increased along with their concentrations. However, simulated atmospheric degradation underestimates summertime TFA deposition five-fold. Continued use of fluorinated compounds is likely to enhance TFA deposition in the future. Additional environmental monitoring and source attribution studies are paramount for refining the assessment of TFA sources and levels for potential health and environmental risks.

## 1 Introduction

Trifluoroacetic acid is the shortest perfluoroalkyl carboxylic acid with the chemical formula $CF_3CO_2H$. It is a very strong acid ($pK_a = 0.52$, Rumble, 2024) and, hence, exists in water only in its deprotonated form (trifluoroacetate, TFA, $CF_3COO$). Commonly, TFA is classified as a per- and polyfluoroalkyl substance (PFAS), for example in the OECD's PFAS-definition (OECD, 2021). Trifluoroacetic acid is completely miscible in water and therefore partitions or dissolves readily into the aqueous phase (surface waters and atmospheric cloud and rain droplets), where it forms salts with other available ions. Because of its high mobility with water, TFA is ubiquitously spread through different environmental media. Although TFA does not generally bioaccumulate, elevated TFA concentrations were, however, observed in leaf material and plant extracts (Rollins et al., 1989; Benesch and Gustin, 2002; Zhang et al., 2019; Scheurer and Nödler,

2021). As other salts, TFA will accumulate in terminal water bodies in arid regions that are dominated by evaporation (Russell et al., 2012; Cahill, 2024) and in the oceans (Scott et al., 2005).

Original toxicology studies of TFA were conducted in the 1990s exploring the impacts on aquatic plants and algae, terrestrial plants, and animals (Boutonnet et al., 1999; Seiber and Cahill, 2021). For the most sensitive aquatic algae (*Selenastrum capricornutum*) a no-observed-effect concentration of $120\,\mu g\,L^{-1}$ was reported by Berends et al. (1999). In 2020, the German environmental protection agency (UBA) derived from toxicological information a non-regulatory guideline value of $60\,\mu g\,L^{-1}$ for TFA in drinking water, emphasizing that the emissions of TFA to water bodies should be minimized and a value of $10\,\mu g\,L^{-1}$ in drinking water should generally not be exceeded (UBA, 2020a, b; Adlunger et al., 2021). In 2023, the Dutch National Institute for Public Health and the Environment (RIVM) derived an indicative drinking water guideline value of $2.2\,\mu g\,L^{-1}$ (RIVM, 2025). Renewed concern was raised about the toxicity of TFA to mammals (Dekant and Dekant, 2023) and the need for more human-relevant studies was expressed (Arp et al., 2024). Currently, the European Food Safety Authority (EFSA) is reviewing the health-based reference values for TFA (EFSA, 2024). In May 2025, the German authorities classified TFA as toxic to reproduction (Category 1B), very persistent and very mobile (vPvM) and submitted a corresponding classification dossier to the European Chemicals Agency (ECHA) for commenting (UBA, 2025). As a consequence of those initiatives, the potential risk posed to human health by TFA is getting increased attention not only by policy makers, but also by the broad public and the media.

Under environmental conditions and in the aqueous phase, TFA is extremely stable (Ellis et al., 2001a) and may have lifetimes in the order of centuries to millennia (Lifongo et al., 2010). Initial studies on TFA degradation under anaerobic conditions in sediment suggested a considerable environmental loss process (Visscher et al., 1994), whereas other studies concluded that TFA is inert (Emptage et al., 1997). In the atmosphere, gaseous trifluoroacetic acid may degrade by reaction with hydroxyl radicals (OH) with lifetimes in the order of 230 d (Hurley et al., 2004) or be subject to overtone-induced photodissociation with tropospheric lifetimes in the order of 8 to 217 years (Reynard and Donaldson, 2002). However, trifluoroacetic acid is more rapidly (order of days) removed from the atmosphere by dry and wet deposition, hence, the OH reaction and photodissociation cannot be seen as significant destruction pathways for TFA in the environment.

A major anthropogenic source of TFA is the atmospheric degradation of volatile fluorinated compounds such as hydrochlorofluorocarbons (HCFCs), saturated hydrofluorocarbons (HFCs), and unsaturated hydrofluoroolefins (HFOs), which are used in a variety of applications, mostly as refrigerants and foam-blowing agents (Solomon et al., 2016;

Madronich et al., 2023). In addition, some fluorinated inhalation anaesthetics (FIAs) are known to produce TFA during their atmospheric degradation (Madronich et al., 2023). Due to their long atmospheric lifetime (order of years) and negative impact on global warming and/or the depletion of the ozone layer, national (e.g., EU Regulation on fluorinated greenhouse gases 2024/573 and American Innovation & Manufacturing (AIM) Act of 2020, European Parliament and Council, 2024; Office of the Law Revision Counsel of the United States House of Representatives, 2020, respectively) and international (Montreal Protocol) regulations are starting to target the emissive use of HCFCs and HFCs. Hence, a transition towards the use of HFOs with short atmospheric lifetimes (order of days to weeks) is ongoing. Some of these newly introduced HFOs degrade with a 100 % molecular yield to TFA in the atmosphere (e.g., HFO-1234yf ($CF_3CH_2F$), Hurley et al., 2008), which is opposed to smaller yields for many of the major HFCs (e.g., 7 %–20 % for HFC-134a ($CF_3CFCH_2$), Wallington et al., 1996). Larger TFA yields and shorter lifetimes will enhance the deposition closer to the major emission sources. A number of atmospheric modelling studies have quantified the effect of the HFC to HFO transition on TFA deposition (Luecken et al., 2010; Henne et al., 2012b; Wang et al., 2018; Holland et al., 2021; Garavagno et al., 2024) and come to the conclusion that TFA concentrations in precipitation will increase by a factor of 10 to 100 in major HFO source regions compared to levels forecasted from HFC degradation or previously observed.

From an observational perspective, considerable increases in TFA concentrations after 2010 for different environmental compartments (precipitation, surface waters, ground water, drinking water) have been reported in various studies (see Arp et al., 2024, and references therein). Although, the general tendency of increased TFA deposition is evident, it is more difficult to draw quantitative conclusions as to sources and fate of TFA, since no continuous monitoring of TFA fluxes in the environment over the complete timescales of the observed increase exists besides a few reported ice core records (Pickard et al., 2020; Hartz et al., 2023).

Besides secondary atmospheric production, other anthropogenic TFA sources to the environment were identified from direct use and discharge from chemical industry and waste water treatment plants (WWTP) (Scheurer et al., 2017). Furthermore, TFA is produced in the environment from the metabolic and ambient degradation of plant protection products (PPP), pharmaceuticals and other per- or polyfluorinated substances carrying a $C-CF_3$-moiety (e.g., Arp et al., 2024, and references therein). The high-temperature use and combustion of fluoropolymers may present an additional anthropogenic source of TFA (Ellis et al., 2001b; Améduri, 2023). Earlier studies have speculated about the existence of natural sources of TFA (i.e., deep-sea vents Scott et al., 2005). More recently (2022/2023), TFA concentrations of ocean samples down to a depth of 4590 m in the North Atlantic were esti-

mated in the range of 237 to 294 ng L$^{-1}$, with slight decreases with depth and higher TFA concentrations at the ocean surface, 260–306 ng L$^{-1}$ (UBA, 2024). A comparison with earlier observations is hindered by questions of measurement quality and validity. Furthermore, it was argued that the large quantities of TFA present in the oceans do not agree with an inventory of anthropogenic consumption of fluorides and potential releases of TFA integrated over the period 1930 to 1999 (Lindley, 2023). However, others concluded against significant natural sources, which would not be able to explain observed increases in TFA levels (Joudan et al., 2021).

There is no general consensus whether or not current and forecasted TFA burdens pose toxicological risks to the environment and/or human health (Madronich et al., 2023; Arp et al., 2024). The UNEP Environmental Effects Assessment Panel reviewed the risks of TFA to ecosystems and human health and conclude that at current TFA concentration levels the risk to humans is de minimis (Neale et al., 2025). Hanson et al. (2024) identified several research gaps that should to be addressed to improve our understanding of the TFA sources, fate and toxicity. Improved atmospheric and hydrological modelling to characterise TFA budgets is one of these areas. Several earlier studies have simulated atmospheric degradation of TFA precursors and TFA deposition from the global to the regional scale, frequently forecasting future burdens of TFA (Kanakidou et al., 1995; Kotamarthi et al., 1998; Luecken et al., 2010; Henne et al., 2012b; Wang et al., 2018; Holland et al., 2021). However, few studies have directly linked observed TFA deposition fluxes to model forecasts and source attribution.

Here, we present an effort to monitor and attribute sources of TFA in precipitation and surface waters in Switzerland by combining three years of continuous observations from a dense network of precipitation and surface water sites with simulations of atmospheric degradation of fluorinated precursor gases. Furthermore, archived precipitation samples dating back to as early as 1984 were analysed for TFA and allow us to consistently document the history and trend of TFA deposition in Switzerland and its relation to atmospheric precursor gases.

## 2 Methods

### 2.1 TFA sampling and analysis

TFA concentrations were continuously observed in different water bodies and precipitation for the period November 2020 to December 2023. Figure 1 and Table 1 give an overview of the location of the observation sites for precipitation and surface water (lakes and rivers). The sampling locations were distributed over all major Swiss river catchments and climatological zones, covering the Ticino river catchment south of the Alps, the Rhone catchment in the central Alps, the Inn catchment in the Eastern Swiss Alps and the Rhine, Thur and Aare catchments in the northern Alps and the Swiss Plateau.

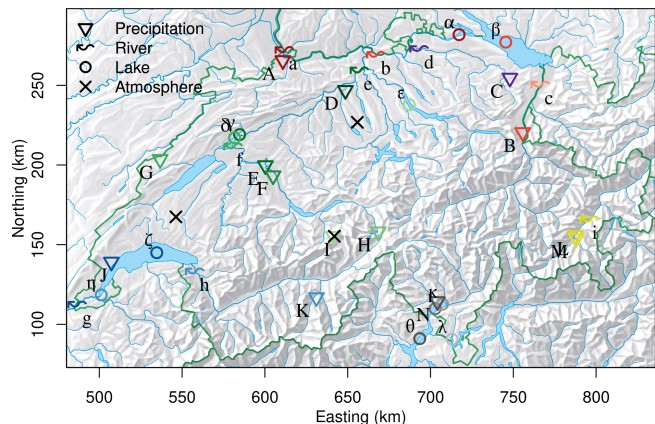

**Figure 1.** Sites with TFA (coloured symbols) and HFO (black crosses) observations in Switzerland. Colours refer to different river basins: Rhine (different shades of red), Rhone (shades of blue), Aare/Limmat (shades of green), Thur (purple), Inn (yellow), Ticino (grey). Precipitation, river and lake sampling sites are marked by triangles/upper case letters, wavy arrows/lower case letters, and open circles/Greek letters, respectively. Letters refer to individual sites as itemised in Table 1. Coordinates refer to Swiss coordinate system (LV03).

TFA samples from precipitation were taken from monthly composite rain samples collected as part of the Swiss 'Observation of Isotopes in the Water Cycle' (ISOT) network (Schürch et al., 2003) and, for the high-Alpine site of Jungfraujoch and the urban site of Bern, by the University of Bern (Climate and Environmental Physics group). Monthly composite samples correspond to the total of all precipitation events, collected continuously. Samplers were not closed during dry periods and therefore, contain dry deposition of atmospheric compounds as well as wet deposition. The sites of the ISOT network are co-located with automatic weather stations of the SwissMetNet operated by the Swiss Federal Office of Meteorology and Climatology (MeteoSwiss). ISOT samples were shipped from the stations to the Swiss Federal Office for the Environment (FOEN) in PE bottles.

Furthermore, TFA concentrations were analysed from surface water samples taken from rivers and lakes, covering the major river network. Flow rate proportional 7d- or 14d composite samples were taken at nine sites of the National Long-Term Network of Swiss Rivers (NAWA FRACHT/NADUF; Storck et al., 2022) and sent to Eawag, Dübendorf, or the International Rhine survey station (RüS), Weil, to compose 28d-composite samples. Grab samples were taken at ten different locations from six of the major Swiss lakes, both at the surface and close to the lake bottom.

Archived precipitation samples, representing monthly integrated precipitation, from the ISOT network for the period 1986 to 2020 and for the sites Jungfraujoch and Bern city (sample archive University of Bern), and archived surface water samples from the NADUF network (Porte-du-

**Table 1.** List of observational sites in Switzerland. Observed TFA concentrations are given as the mean and standard deviation ($1\sigma$) for $N$ samples in the the period 2021–2023. Deposition flux, river load and pooled TFA in lakes, were derived by multiplication with precipitation rates, flow rates, and lake volumes, respectively. Sites were associated with major Swiss river basins. The rivers Aare, Limmat and Thur all empty into the Rhine prior to the Rhine exiting Switzerland at Weil. Lake samples where taken both at surface and bottom of the lake, if not otherwise stated. Labels as used in Fig. 1.

| Site | Label | Basin | Longitude °E | Latitude °N | Altitude m a.s.l. | Conc. µg L$^{-1}$ | Flux/load | $N$ |
|------|-------|-------|-----------|----------|---------|------|-----------|---|
| Precipitation | | | | | | | kg km$^{-2}$ yr$^{-1}$ | |
| Basel-Binnigen (BAS) | A | Rhine | 7.5827 | 47.5412 | 316 | $0.73 \pm 0.15$ | $0.61 \pm 0.17$ | 36 |
| Sevelen (SEV) | B | Rhine | 9.4907 | 47.1165 | 460 | $0.47 \pm 0.10$ | $0.59 \pm 0.18$ | 36 |
| St.Gallen (STG)[a] | C | Thur | 9.3995 | 47.4256 | 779 | $0.43 \pm 0.11$ | $0.70 \pm 0.23$ | 35 |
| Buchs-Suhr (BUS) | D | Aare | 8.0820 | 47.3710 | 397 | $0.51 \pm 0.12$ | $0.50 \pm 0.16$ | 36 |
| Bern (BER) | E | Aare | 7.4432 | 46.9495 | 553 | $0.96 \pm 0.35$ | $0.74 \pm 0.21$ | 35 |
| Belp (BEP) | F | Aare | 7.5035 | 46.8910 | 520 | $0.57 \pm 0.17$ | $0.49 \pm 0.16$ | 36 |
| La Brévine (BRL) | G | Aare | 6.6071 | 46.9801 | 1042 | $0.63 \pm 0.20$ | $0.66 \pm 0.19$ | 27 |
| Grimsel – Hospiz (GRH) | H | Aare | 8.3316 | 46.5712 | 1950 | $0.44 \pm 0.13$ | $0.63 \pm 0.16$ | 36 |
| Jungfraujoch (JFJ) | I | Aare | 7.9800 | 46.5489 | 3570 | $0.30 \pm 0.08$ | $0.31 \pm 0.10$ | 35 |
| Nyon – Changins (CGI) | J | Rhone | 6.2330 | 46.3974 | 436 | $0.66 \pm 0.21$ | $0.46 \pm 0.14$ | 34 |
| Grächen – St. Niklaus (GRC) | K | Rhone | 7.8411 | 46.2024 | 1605 | $0.86 \pm 0.22$ | $0.41 \pm 0.12$ | 35 |
| Samedan (SAM)[b] | L | Inn | 9.8808 | 46.5275 | 1709 | $0.64 \pm 0.29$ | $0.52 \pm 0.26$ | 16 |
| Pontresina (PON)[b] | M | Inn | 9.8878 | 46.5094 | 1790 | $0.93 \pm 0.74$ | $0.41 \pm 0.26$ | 11 |
| Locarno-Monti (OTL) | N | Ticino | 8.7874 | 46.1726 | 379 | $0.65 \pm 0.16$ | $1.28 \pm 0.44$ | 33 |
| Rivers | | | | | | | Mg yr$^{-1}$ | |
| Weil | a | Rhine | 7.5947 | 47.6014 | 244 | $0.65 \pm 0.1$ | $19.9 \pm 3.19$ | 39 |
| Rekingen | b | Rhine | 8.3294 | 47.5708 | 326 | $0.47 \pm 0.08$ | $6.18 \pm 0.99$ | 39 |
| Diepoldsau | c | Rhine | 9.6409 | 47.3831 | 410 | $0.36 \pm 0.06$ | $2.47 \pm 0.40$ | 39 |
| Andelfingen | d | Thur | 8.6767 | 47.5996 | 361 | $0.81 \pm 0.13$ | $1.26 \pm 0.20$ | 39 |
| Brugg | e | Aare | 8.1949 | 47.4825 | 332 | $0.55 \pm 0.09$ | $5.28 \pm 0.85$ | 39 |
| Hagneck | f | Aare | 7.1842 | 47.0555 | 441 | $0.41 \pm 0.07$ | $2.17 \pm 0.35$ | 39 |
| Chancy | g | Rhone | 5.9707 | 46.1530 | 336 | $0.88 \pm 0.15$ | $8.37 \pm 1.41$ | 35 |
| Porte du Scex | h | Rhone | 6.8886 | 46.3496 | 377 | $0.80 \pm 0.13$ | $4.02 \pm 0.64$ | 39 |
| S-chanf | i | Inn | 9.9952 | 46.6157 | 1650 | $0.33 \pm 0.05$ | $0.19 \pm 0.03$ | 39 |
| Lakes | | | | | | | Mg | |
| Lake Constance, Berlingen | $\alpha$ | Rhine | 9.0000 | 47.6763 | 397 | $0.35 \pm 0.14$ | $16.99 \pm 6.94$ | 6 |
| Lake Constance, Uttwil | $\beta$ | Rhine | 9.3739 | 47.6286 | 397 | $0.34 \pm 0.14$ | $16.26 \pm 6.64$ | 6 |
| Lake Biel, Niedau[c] | $\gamma$ | Aare | 7.2371 | 47.1227 | 429 | $0.51 \pm 0.26$ | $0.57 \pm 0.28$ | 4 |
| Lake Biel, DW4[d] | $\delta$ | Aare | 7.1976 | 47.1043 | 429 | $0.56 \pm 0.28$ | $0.62 \pm 0.31$ | 4 |
| Lake Zurich, Thalwil | $\epsilon$ | Limmat | 8.5910 | 47.2858 | 406 | $0.34 \pm 0.12$ | $1.14 \pm 0.40$ | 8 |
| Lake Geneva, SHL2[d] | $\zeta$ | Rhone | 6.5887 | 46.4527 | 372 | $1.00 \pm 0.50$ | $88.61 \pm 44.3$ | 4 |
| Lake Geneva, Paquis[c] | $\eta$ | Rhone | 6.1578 | 46.2102 | 372 | $0.92 \pm 0.65$ | $81.93 \pm 57.9$ | 2 |
| Lake Maggiore, Ghiffa[d] | $\iota$ | Ticino | 8.6448 | 45.9641 | 193 | $0.25 \pm 0.12$ | $9.49 \pm 4.74$ | 4 |
| Lake Maggiore, Ascona[c] | $\kappa$ | Ticino | 8.7688 | 46.1319 | 193 | $0.43 \pm 0.22$ | $16.42 \pm 8.21$ | 4 |
| Lake Maggiore, Locarno[c] | $\theta$ | Ticino | 8.8270 | 46.1570 | 193 | $0.46 \pm 0.23$ | $17.65 \pm 8.83$ | 4 |
| Lake Lugano, Figino | $\lambda$ | Ticino | 8.8968 | 45.9584 | 271 | $0.54 \pm 0.19$ | $2.53 \pm 0.89$ | 8 |

[a] The precipitation site St. Gallen was associated with the Thur basin although it is at the border between the Thur and Rhine basin. [b] Sampling in Pontresina only until March 2022. Since August 2022 replaced by sampling in nearby Samedan. [c] Sampling at lake surface. [d] Sampling at lake bottom.

Scex/Rhone, Weil/Rhine) were analysed for TFA. Archived samples were stored in amber glass bottles capped with screw tops and inserted septa at temperatures of 12 to 14 °C (Schotterer et al., 2010) under predominantly dark conditions in a cellar. Storage conditions were checked for long-term sta-

bility of stable water isotope consistency by redetermination of stable isotopes using Cavity Ring Down Spectrometry after decades of storage (Leuenberger and Ranjan, 2021) in comparison to previous conventional determination by mass spectrometry on these samples right after sampling as part of

the IAEA GNIP programme (International Atomic Energy Agency, 2025). Unused new septa were eluated in the lab with blank-free water and tested for TFA release to assure that storage did not change TFA concentrations.

Finally, four water samples from the ice core B82-1, drilled in 1982 at the saddle of Colle Gnifetti glacier (45.92997° N, 7.87436° E) at an elevation of 4450 m were analysed for TFA. These four samples correspond approximately to precipitation originating from 1978 (Bag 8), 1942 (Bag 50) and 1892 (Bag 97, two samples) (Döscher et al., 1995, 1996).

All water samples were filled in PE centrifuge vials prepared at FOEN, RüS or Eawag and shipped to the DVGW-Technologiezentrum Wasser (German Water Centre) in Karlsruhe, Germany, where they were analysed by ion exchange liquid chromatography (LC) coupled to electrospray tandem mass spectrometry (MS-MS) with a limit of quantification (LOQ) determined as 50 ng L$^{-1}$ (Scheurer et al., 2017). Any values reported below the LOQ (97 of a total of 1570 samples) were given as 0.04 ng L$^{-1}$ and were treated as such in any averaging. Field blank tests were conducted by filling and shipping blank-free ultra-pure water in the same way as the samples.

## 2.2 Monitoring of atmospheric precursors

The global monitoring of a number of atmospheric precursor gases of TFA (HCFCs, HFCs, FIAs) is routinely conducted by two principal networks: (1) the Advanced Global Atmospheric Gases Experiment (AGAGE) network of continuous observation sites (Prinn et al., 2018) and (2) the National Oceanic And Atmospheric Administration (NOAA) flask sampling network, which collects weekly air flasks from a number of sites worldwide and subsequently analyses these flask in a central laboratory (Montzka et al., 2018). As part of the AGAGE network Empa operates a Medusa GC/MS (Miller et al., 2008) at the high-altitude site Jungfraujoch in the Central Swiss Alps (Reimann et al., 2020). This includes the continuous analysis of newly-marketed HFOs such as HFO-1234yf, HFO-1234ze(E), HCFO-1233zd(E) (Vollmer et al., 2015), and HFO-1336mzz(Z) (Rust et al., 2023). In addition, observations from two Swiss lower altitude locations are available from two multi-month campaigns conducted at Beromünster and Sottens as well as from a site in The Netherlands, Cabauw, (Rust et al., 2022, 2023). Additional long-term European monitoring sites for halogenated compounds include Mace Head, Ireland, Tacolneston, UK, and Monte Cimone, Italy, which are part of the AGAGE network.

## 2.3 Simulation of atmospheric degradation of fluorinated precursors and TFA deposition

### 2.3.1 Long-lived HCFCs, HFCs, fluorinated inhalation anaesthetics

TFA deposition contributions from long-lived HFCs, HCFCs and, FIAs were calculated based on (i) a single-box model presented in Behringer et al. (2021) and (ii) using the AGAGE atmospheric 12-box model (Rigby et al., 2008), both constrained by atmospheric concentration observations from the AGAGE network (Prinn et al., 2018). The first approach was used for all compounds listed in Table 2, whereas the second could not be applied to HCFC-123, HCFC-133a and the fluorinated inhalation anaesthetics for which either no reliable, long-term atmospheric record or no recent 12-box simulations were available at the writing of this manuscript. Other widely-used HCFCs, like HCFC-22, HCFC-141b, and HCFC-142b, do not contain a CF$_3$ group and, hence, do not contribute to TFA formation in the atmosphere and were not considered here. Results from both approaches are compared to assure consistency.

Both calculations assume that the main loss of HCFCs, HFCs, and FIAs is through reaction with OH. After the initial reaction with OH a constant molecular yield to TFA was assumed for each compound (Table 2). In both approaches, it is assumed that TFA deposits close to where it was formed in the atmosphere, such that molecular loss rates per surface area multiplied by molecular TFA yields directly provide molecular TFA deposition rates. Given the relatively homogeneous atmospheric distributions of the precursor gases this assumption should give a good first order estimate of average hemispheric deposition rates. This expectation is in line with previous model studies of HCFC/HFC degradation, which suggest relatively homogeneous TFA spatial distributions (Kanakidou et al., 1995; Kotamarthi et al., 1998; Holland et al., 2021). Spatial differences may occur due to variability in OH and precipitation distributions.

For the single-box model the atmospheric lifetime and background mole fractions observed at the Jungfraujoch observatory, both listed in Table 2, are the only drivers determining the loss rate $L_A$ (units mol m$^{-2}$ yr$^{-1}$) of a compound $A$

$$L_A = \frac{C_A}{\tau_A} = \frac{N \chi_A}{\tau_A}, \tag{1}$$

where $C_A$ is the column molar density of compound $A$ (units mol m$^{-2}$), $\tau_A$ is the OH lifetime (years), $N$ the column molar density of air, and $\chi_A$ the average mole fraction of compound $A$. A single deposition rate for each compound representative of the years 2021–2023 was calculated. In the second approach, monthly variable OH concentrations for each of the 12 atmospheric boxes were used to more explicitly calculate monthly precursor loss rates for each of the 12 boxes. We combine the loss rates from all six northern hemispheric boxes to derive a loss rate per surface area, which, once

**Table 2.** HFCs, HCFCs, FIAs considered for calculation of TFA formation/deposition. Average mole fractions as observed at the AGAGE site Jungfraujoch, Switzerland, during background conditions in 2021–2023, except for HCFC-123, which was taken from observations from the NOAA flask network (Montzka et al., 2018) (Isacc Vimont, NOAA, personal communication, TS3). TFA yields are taken from (Madronich et al., 2023) and references therein. Mean (2021–2023) deposition rates are taken from the 12-box model, if not stated otherwise, and calculated with mean, minimal, and maximal TFA yields and represent average northern hemispheric conditions.

| Compound | Mole fraction (pmol mol$^{-1}$) | Lifetime (years) | Molecular TFA yield (–) | Deposition rate (g km$^{-2}$ yr$^{-1}$) |
|---|---|---|---|---|
| hydrochlorofluorocarbons | | | | |
| HCFC-123* | 0.325 | 1.3 | 0.6 (0.5–0.7) | 5.4 (4.5–6.3) |
| HCFC-124 | 0.921 | 5.9 | 1.0 | 5.7 |
| HCFC-133a* | 0.528 | 4.6 | 0.39 (0.22–0.55) | 1.6 (0.91–2.3) |
| hydrofluorocarbons | | | | |
| HFC-125 | 45.7 | 30.0 | 0.06 (0.01–0.1) | 2.9 (0.52–5.2) |
| HFC-134a | 134.5 | 13.4 | 0.14 (0.07–0.2) | 49 (25–72) |
| HFC-143a | 31.38 | 47.1 | 0.16 (0.02–0.3) | 3.5 (0.44–6.6) |
| HFC-227ea | 2.396 | 38.9 | 1.0 | 2.4 |
| HFC-236fa | 0.257 | 213 | 0.2 (0.1–0.3) | 0.007 (0.0035–0.011) |
| HFC-245fa | 3.964 | 7.7 | 0.09 (0.01–0.17) | 1.6 (0.18–3.1) |
| HFC-365mfc | 1.323 | 8.7 | 0.55 (0.1–1.0) | 3.0 (0.54–5.4) |
| HFC-43-10mee | 0.323 | 17.0 | 0.57 (0.54–0.6) | 0.39 (0.37–0.41) |
| anaesthetics | | | | |
| halothane* | 0.010 | 1.0 | 0.6 (0.5–0.7) | 0.22 (0.18–0.25) |
| isoflurane* | 0.152 | 3.2 | 0.95 (0.92–0.98) | 1.6 (1.6–1.7) |
| desflurane* | 0.345 | 9.0 | 0.12 (0.03–0.2) | 0.16 (0.042–0.28) |
| sevoflurane* | 0.264 | 1.4 | 0.49 (0.02–0.95) | 3.3 (0.14–6.5) |
| Total | | | | 81 (43–119) |

* Derived from single-box model and NH background concentrations.

again, can be multiplied by molecular TFA yields to provide mean hemispheric deposition rates. The precursor emissions entering the model were constrained by using the temporal evolution of atmospheric observations from different sites of the AGAGE network representative of the mole fraction in each of the four lower atmospheric boxes in an inverse estimation step. The approach yields monthly deposition rates from 1995 to 2023. The uncertainty on the estimated average hemispheric deposition rate will mainly be driven by the uncertainty of the emissions. The latter was estimated as part of the 12-box model simulations and, as an a posteriori estimate of a Bayesian inversion, includes contributions from the observational uncertainty and transport model uncertainty. Furthermore, uncertainties of the OH lifetime are considered. For details on the 12-box model please refer to Rigby et al. (2008).

Three scenarios were calculated, assuming the average, minimal and maximal TFA yield as suggested by Madronich et al. (2023) and given in Table 2. The latter two serve as an uncertainty estimate and assume fully correlated uncertainty between TFA yields for different compounds. For compounds with a commonly agreed on TFA yield of 1, this value was used in all three scenarios.

### 2.3.2 Short-lived HFOs

Due to the short lifetimes of HFOs, a simple box-model approach, as for the long-lived compounds, cannot be applied, but explicit transport and chemistry of emitted HFOs needs to be considered. As a replacement refrigerant in mobile and stationary air conditioning, HFO-1234yf is currently the most widely used HFO in Europe and exhibits a 100 % molecular TFA yield (Hurley et al., 2008). Hence, we explicitly simulated TFA formation and deposition from HFO-1234yf emissions, whereas TFA from other HFOs was only considered indirectly.

To this extent, we utilised a similar model approach as in Henne et al. (2012b), based on the Lagrangian Particle Dispersion Model (LPDM) FLEXPART (Pisso et al., 2019), which offers the possibility to cover a sufficiently large transport domain at relatively low computational costs. This allowed the description of HFO-1234yf degradation to TFA, which happens in the order of days to weeks. The model simulates atmospheric HFO-1234yf mole fractions based on emissions from Europe only. HFO-1234yf degradation due to reaction with OH and Cl is described following the scheme and rate constants

$(k_{OH} = 1.26 \times 10^{-12} \exp(-35/T) \, \mathrm{cm^3 \, molec.^{-1} \, s^{-1}}$ and $k_{Cl} = 7.03 \times 10^{-11} \, \mathrm{cm^3 \, molec.^{-1} \, s-1})$ suggested by Hurley et al. (2008). OH and Cl concentrations were fixed to a constant climatology in FLEXPART utilising a multi-model OH estimate (Fiore et al., 2009). The resulting trifluoroacetyl fluoride (TFF, $CF_3C(O)F$) further hydrolyses in cloud droplets, forming trifluoroacetic acid (Luecken et al., 2010). In FLEXPART, hydrolysis and deprotonation happens instantaneously during a single model time step of 600 s as soon as any cloud water is present in a respective grid cell. TFA is then removed from the atmosphere through dry and wet deposition. Both processes are represented in FLEXPART, assuming that TFA behaves like nitric acid (employing a constant effective Henry's Law constant of $1 \times 10^{14} \, \mathrm{M \, atm^{-1}}$; Wesely, 1989; Stohl et al., 2005), which corresponds to the vast majority of TFA residing in the liquid phase and, hence, fast washout through in-cloud wet scavenging. FLEXPART treats in-cloud and below-cloud scavenging of gases separately, with the latter only depending on precipitation rates. Additionally, gas phase destruction of trifluoroacetic acid by reaction with OH is considered in the model (Hurley et al., 2008). Recently, Holland et al. (2021) pointed out the importance of stabilised Criegee intermediates in gas phase trifluoroacetic acid destruction. However, they showed that including this degradation path was mainly important in tropical forested regions and, hence, was omitted in our simulations.

As a Lagrangian particle dispersion model, FLEXPART does not forecast concentrations in a spatially-fixed grid, but along a multitude of trajectories of the flow field assuming a Markov process to describe turbulent transport. Trajectories were calculated for air parcels released at the model's surface and representing a given amount of emitted HFO-1234yf (see below). Individual air parcels were tracked for 60 d in the atmosphere undergoing mean transport and turbulent and convective dispersion. The chemical composition of these air parcels was simulated according to the above description. Gridded concentrations and deposition rates were then obtained by sampling all air parcels within a given grid cell or column and during a respective averaging interval.

The FLEXPART model version, previously used by Henne et al. (2012b) for the simulations of HFO-1234yf degradation, branched off from the official FLEXPART release version 8.1. The developments required for the description of TFA formation were not fed back into the main development version of FLEXPART. The previous version, however, was lacking several important developments that were implemented in FLEXPART in the meantime (especially parallelisation and improved handling of input/output). Hence, the features required for a description of TFA formation and deposition were ported to a more recent version of FLEXPART (9.02), allowing for higher-resolution simulations, larger problem sizes, and accelerated calculations.

For the present study FLEXPART was driven by meteorological fields taken from the European Centre for Medium range Weather Forecast (ECMWF) making use of their HRES analysis/forecast system and made available for nested input to FLEXPART. We used a high-resolution domain ($0.1° \times 0.1°$; 8 km $\times$ 10 km) for most of Western and Central Europe with hourly resolution. Outside this domain, ECMWF fields were available at $0.5° \times 0.5°$ and 3-hourly resolution. In comparison to our previous study (Henne et al., 2012b), we increased the number of model air parcels released during a month from 1.5 million to 10 million, which results in a more robust estimation of concentration fields. The maximal age of air parcels was increased as well, from 30 to 60 d, which assures that also during the cold season most of the released HFO-1234yf will be degraded before individual air parcels are terminated.

Currently, no European emission inventory or database for HFOs exists. To realistically drive our degradation simulations we used annual mean HFO-1234yf emissions derived from concentration observations at Jungfraujoch, Mace Head, Monte Cimone, and Tacolneston and inverse modelling (Vollmer et al., 2025). Here, we only employ the estimates as obtained from Empa's Lagrangian regional inversion system (ELRIS) (Henne et al., 2016). These consist of a total of four sensitivity inversions obtained by employing two different a priori distributions (population-based and uniform over land) as well as two different atmospheric transport models (FLEXPART and NAME). HFO-1234yf emissions from EU27+ (EU27 plus Norway, Switzerland, United Kingdom) increased from virtually none in 2014 to about $2.5 \pm 0.3 \, \mathrm{Gg \, yr^{-1}}$ in 2023 (Fig. A1). The inversion assigned large HFO-1234yf emissions to the Benelux region, UK, Western and South-eastern Germany, as well as France and the Po Valley. This distribution largely follows the expectation that HFO-1234yf emissions will align with population distributions. To overcome limitations in terms of spatial distribution and resolution of the inversion results, we re-distributed the estimated European emission total at $0.1° \times 0.1°$ spatial resolution according to proxy distributions previously used by Henne et al. (2012b) for generating HFO-1234yf emission maps. These include a proxy for emissions along the road transportation network, as expected for a compound dominantly used in mobile air conditioning, and population density as a proxy for both mobile and stationary HFO-1234yf applications (Fig. A2). Remapped annual average emissions were then used to create monthly release files for FLEXPART by linearly interpolating between annual averages. No seasonality in emissions was considered.

The model simulations were initialised for 1 October 2020 TS4 and we allowed for three months of model spin-up. Hence, simulation results were available from 1 January 2021 onwards. Daily mean concentration fields of HFO-1234yf, TFF and TFA, as well as dry and wet deposition rates of TFA were stored at $0.1° \times 0.1°$ resolution.

In order to access additional aspects of the TFA deposition and evaluate uncertainties in the chemistry and transport simulations, sensitivity simulations were carried out next to

the base simulation (BASE). One experiment contained only HFO-1234yf emissions from Switzerland (CH). Since we only consider first-order degradation chemistry, subtracting mole fractions and deposition rates obtained from this experiment from the base simulation with complete European emissions, one can obtain the European contribution to concentrations and deposition rates in Switzerland. Two additional sensitivity simulations (SLOW and FAST) were run to evaluate the impact of the uncertainty of the rate constant of the initial degradation reaction of HFO-1234yf with OH. Rates were varied to the lower/upper range given by Papadimitriou et al. (2008), which reflects an overall 10 % slower/faster initial reaction. The impact of the initial degradation rate on TFA deposition is not expected to be linear and will depend on the relative sampling location with respect to the main emission sources. The hydrolysis of TFF to TFA was considered to be complete within a single model time step of 600 s (compared to a hydrolysis rate of $150\,\mathrm{s}^{-1}$ according to George et al., 1994), too short for the current model scale to allow for quantitative sensitivity tests and to assess it as a source of uncertainty.

The final deposition of TFA from the atmosphere is described through three different pathways in the model: dry deposition at the ground, wet deposition through below-cloud scavenging and wet deposition through in-cloud scavenging. FLEXPART employs parameterisations for dry deposition and in-cloud scavenging that depend on the Henry's law constant of the gas, whereas below-cloud scavenging only depends on the precipitation intensity. Using a very large Henry's law constant in the model, as suggested for $HNO_3$, assumes that almost all TFA will reside in the cloud-phase, and, in case of precipitating clouds, resulted in very large scavenging coefficients (i.e., very rapid ($< 1\,\mathrm{h}$) washout time scales) in our BASE simulations. A sensitivity simulation with reduced below-cloud scavenging coefficients resulted in an increased TFA lifetime against washout. However, the results of these simulations were considered unrealistic since, on the one hand, they reduced TFA wet deposition, but, on the other hand, increased TFA dry deposition to such extent that dry deposition dominated over wet deposition. The latter seems to be unrealistic given previous simulation results using atmospheric transport and chemistry models other than FLEXPART (Luecken et al., 2010; Wang et al., 2018; Henne et al., 2012b; Holland et al., 2021). The latest version of FLEXPART (v11, Bakels et al., 2024) features an improved description of atmospheric deposition and future simulations should benefit from this update and should also consider a more specific and temperature dependent Henry's law constant for TFA.

HFOs/HCFOs other than HFO-1234yf that are already marketed at considerable amounts and detectable in the atmosphere are HFO-1234ze(E), HCFO-1233zd(E) (Vollmer et al., 2015), and HFO-1336mzz(Z) (Rust et al., 2023). HFO-1234ze(E) has a similar atmospheric lifetime as HFO-1234yf (approximately 19 vs. 12 d). Therefore, and assuming a sim-

ilar European source distribution, the TFA deposition distribution resulting from HFO-1234ze(E) is expected to be similar to that of HFO-1234yf. However, the molecular TFA yield from HFO-1234ze(E) is expected to be considerably smaller than for HFO-1234yf, 2 %–30 % (Madronich et al., 2023). At the same time, current (2021-2023) European emissions of HFO-1234ze(E) are about a factor of two smaller than those of HFO-1234yf (see Table 3). Hence, an additional TFA contribution from HFO-1234ze(E) in the order of 1 %–15 % of that of HFO-1234yf can be expected. The atmospheric lifetime of HFO-1336mzz(Z) is more than twice that of HFO-1234yf. Hence, resulting TFA production and deposition is expected to be more dispersed. No European scale emission estimate of HFO-1336mzz(Z) is available. Comparing the emission estimate in Rust et al. (2023) for the Netherlands with a Dutch HFO-1234yf estimation (Vollmer et al., 2025), 30 vs. $120\,\mathrm{Mg\,yr}^{-1}$, we assume European emissions of HFO-1336mzz(Z) to be four times smaller (mass basis) than HFO-1234yf. Considering a molecular TFA yield of 4 %–60 % (Madronich et al., 2023), an upper range of TFA deposition from HFO-1336mzz(Z) can be expected in the order of 1 %–15 % of that of HFO-1234yf. HCFO-1233zd(E) has an atmospheric lifetime of 42 d and an estimated molecular TFA yield of 2 %–30 % (Madronich et al., 2023). With European mass emissions being about a factor of two smaller than those of HFO-1234yf (Table 3) associated TFA deposition is expected in the range of 1 %–15 % of that of HFO-1234yf. Overall, TFA deposition from these compounds can be expected to add about 24 % to that of HFO-1234yf, with this value being the mean considering the literature on TFA yields (minimum–maximum: 3 %–45 %). In all estimates of TFA depostion from HFO-1234yf, we add these 24 % to account for other HFOs in use, well knowing that due to different atmospheric lifetimes the relationship between emissions and TFA deposition will not be fully linear everywhere.

## 3   Results

### 3.1   TFA concentrations in recent precipitation and surface waters

In total, 441 monthly precipitation samples were analysed for the period January 2021 to December 2023. TFA concentrations ranged from $0.04\,\mathrm{\mu g\,L}^{-1}$ at Buchs-Suhr in January 2021 to $5.7\,\mathrm{\mu g\,L}^{-1}$ at Bern in June 2023. Note that the lowest concentration was just below the estimated limit of quantification of $0.05\,\mathrm{\mu g\,L}^{-1}$. Overall, only three precipitation samples fell below this limit. The average and standard deviation of all samples was $0.58 \pm 0.58\,\mathrm{\mu g\,L}^{-1}$, indicating the large variability in the observations.

The level of observed TFA concentrations in rainwater was strongly dependent on the season, with generally higher concentrations during the summer and lower concentrations during the winter (Fig. 2a). Some exceptions for individ-

**Table 3.** HFOs/HCFOs considered for calculation of TFA formation/deposition. Background mole fractions as observed at AGAGE site Jungfraujoch, Switzerland. TFA yields are taken from (Madronich et al., 2023) and references therein. EU27+ emission estimates are taken from inverse modelling for the years 2021–2023 and using the ELRIS inversion system (Vollmer et al., 2025), if not stated otherwise.

| Compound | Mole fraction (pmol mol$^{-1}$) | Lifetime (days) | Molecular TFA yield (–) | EU27+ emissions (Gg yr$^{-1}$) |
|---|---|---|---|---|
| hydrofluoroolefins | | | | |
| HFO-1234yf | 0.060 | 12 | 1 | $2.0 \pm 0.23$ |
| HFO-1234ze(E) | 0.095 | 14 | 0.16 (0.02–0.3) | $1.3 \pm 0.07$ |
| HFO-1336mzz(Z) | 0.02 | 27 | 0.32 (0.04–0.6) | 0.6* |
| hydrochlorofluoroolefins | | | | |
| HCFO-1233zd(E) | 0.200 | 42 | 0.16 (0.02–0.3) | $1.2 \pm 0.07$ |

* Upscaled from Dutch HFO-1336mzz(Z) estimate of Rust et al. (2023).

ual months (e.g., February 2023) exist. The variability between sites was smaller during the winter, whereas during the summer individual sites frequently differed considerably from each other. Furthermore, observed concentration distributions varied from site to site with the lowest concentrations generally observed at the inner Alpine site of Jungfraujoch (JFJ), and highest concentrations at the city location of Bern (BER, Fig. 2a).

High TFA concentrations are not necessarily connected with large TFA deposition loads, but may arise from small precipitation amounts as well. Observed TFA concentrations in precipitation were converted into TFA deposition rates using monthly precipitation amounts. TFA deposition rates shared the same seasonal features as the TFA concentrations with largely increased rates during the summer months (Fig. 2b). Furthermore, differences between river basins are more clearly discernible during the summer. By far the largest average TFA deposition rates were observed south of the Alps at Locarno (OTL, 1.28 kg km$^{-2}$ yr$^{-1}$), whereas deposition rates in the central Alps (Jungfraujoch, JFJ, 0.31 kg km$^{-2}$ yr$^{-1}$, and Grächen St. Niklaus, GRC, 0.41 kg km$^{-2}$ yr$^{-1}$) were smallest. The average TFA deposition rate across all sites was $0.59 \pm 0.23$ kg km$^{-2}$ yr$^{-1}$ for the years 2021 to 2023 (here, we explicitly exclude the samples taken in late 2020 to rule out seasonal effects in the calculation of the average).

The pronounced seasonality in TFA rainwater concentrations and TFA deposition rates, together with the much weaker seasonality in TFA precursor concentrations (see Fig. 5), are a strong indicator that atmospheric degradation processes are the main driver of atmospheric TFA inputs: degradation of precursors is generally accelerated during the summer due to larger availability of radical species and higher temperatures. Other direct and seasonally independent TFA sources to the atmosphere (e.g., from waste incineration or industry) do not seem to contribute a major fraction at most sites, given the current observational data.

Continuous river water monitoring resulted in the analysis of 347 28 d samples from nine locations. In these, concentrations ranged from 0.21 to 2.8 µg L$^{-1}$, measured in August 2022 and February 2021 at Porte du Scex (at the exit of the Rhone valley into Lake Geneva), respectively. Despite the large seasonality in TFA concentrations in rainwater, there is very little evidence for a robust seasonality in TFA concentrations in the investigated rivers (Fig. 3a). However, seasonality of TFA loads (concentration multiplied by flow rate) with a summertime peak was observed at S-Chanf and Diepoldsau, where there are no greater lakes in the catchment upstream and we can expect little delay between the concentration maximum in precipitation and that in river discharge (Fig. 3b). The latter seasonality of loads is clearly related to the discharge seasonality at these sites, with greater discharge in the summer, partly from snow and glacier melt, offsetting the concentration signal. An inverted seasonality in TFA concentrations was observed for some, more polluted sites, most prominently for Porte du Scex. Here, largely increased TFA concentrations were observed during the spring months. These increases are most likely due to reduced dilution of industrial emissions or communal WWTP contributions and are aligned with a minimum in the Rhone flow rate, such that the TFA load remains more constant throughout the whole year. In winter, Rhone discharge at Porte du Scex consists almost completely of stored water from the previous summer, which can be deduced from stable isotope signature at Porte-du-Scex (ISOT) and general seasonal precipitation patterns (Schotterer et al., 2010), together with discharge data (FOEN) and storage data of hydropower dams (Swiss Federal Office of Energy, 2025). Therefore, the observed peak in TFA concentrations during winter/early spring is most likely due to the large share of summer precipitation stored in hydroelectric dams and released in winter.

A clear increase of TFA concentrations along the individual rivers was noticeable. Concentrations were smallest for locations higher up in the individual catchments (Diepoldsau, Hagneck, S-chanf) than at the outlets of the Rhone (Chancy)

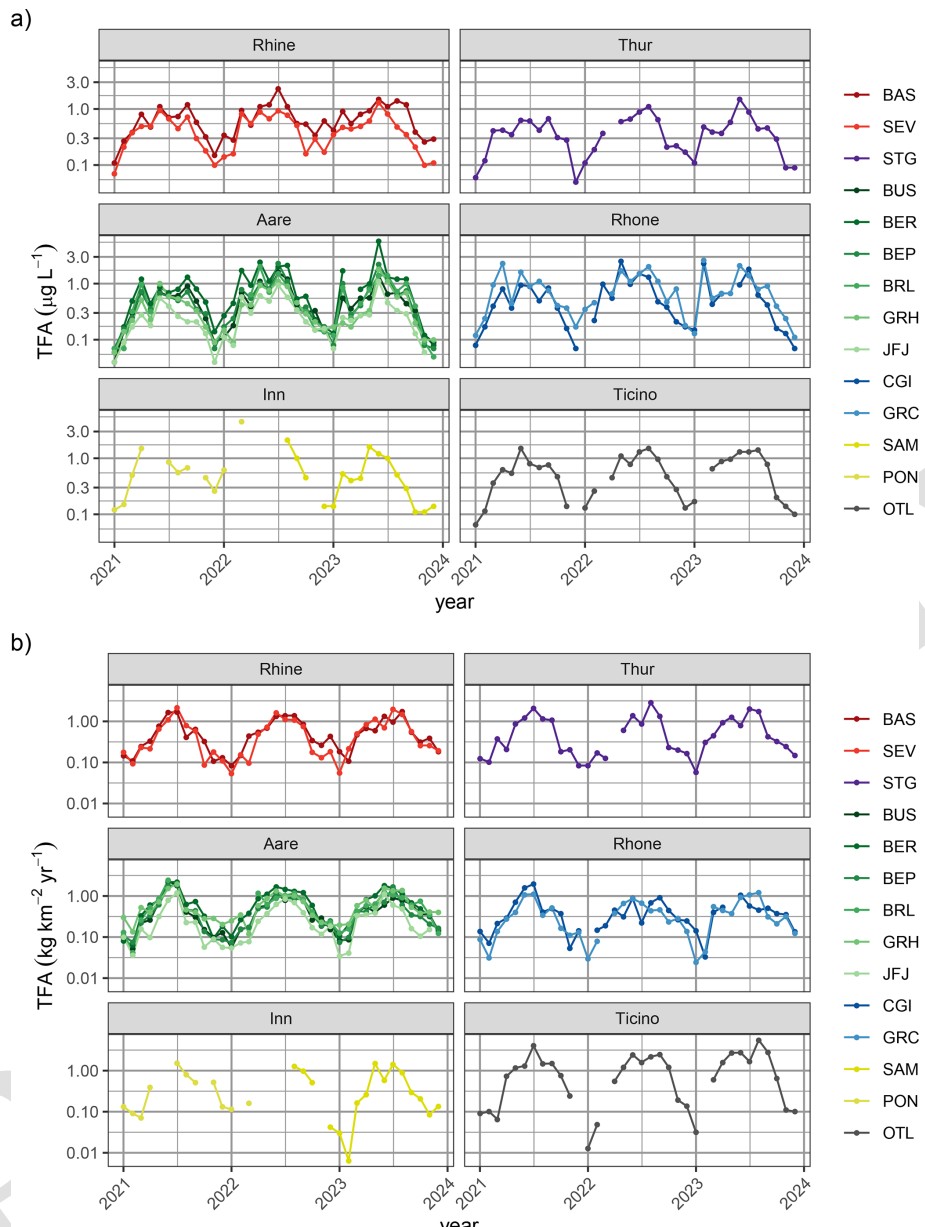

**Figure 2.** Time series of **(a)** measured TFA concentrations in monthly rainwater samples and **(b)** derived monthly TFA deposition rates. Sites are coloured by different catchments: Rhine (red), Thur (purple), Aare (green), Rhone (blue), Inn (yellow), Ticino (grey). Darker colours represent sites farther down-stream in the respective catchment. Note logarithmic *y*-axis scale.

and Rhine (Weil). Increased TFA levels were also observed at the outlet of the Thur into the Rhine (Andelfingen). Samples taken relatively high within the Inn and Rhine catchments (e.g., S-chanf or Diepoldsau) showed generally lower concentrations than precipitation in the respective areas (e.g., Pontresina/Samedan, Sevelen), suggesting that increases in rainwater concentrations have not reached the groundwater sources of these areas and/or that glacial melt and runoff contributes considerably to diluting TFA from rainwater. More positive stable isotopes signature $\delta^{18}$O from sum-

mer/fall precipitation can be seen in the Rhone in winter, which supports this assumption (Schotterer et al., 2010). Additional surface water grab samples from Massa river at station Blatten (Naters) below the Aletsch glacier between May and October 2022 exhibited TFA concentrations of 0.15 to 0.22 µg L$^{-1}$, reinforcing the findings of low TFA concentrations in remote areas and dilution with melt water, similar to Diepoldsau and S-Chanf. This is also supported by glacial ice samples taken from ice cores extracted from the Colle Gnifetti glacier in the upper Rhone catchment that did not

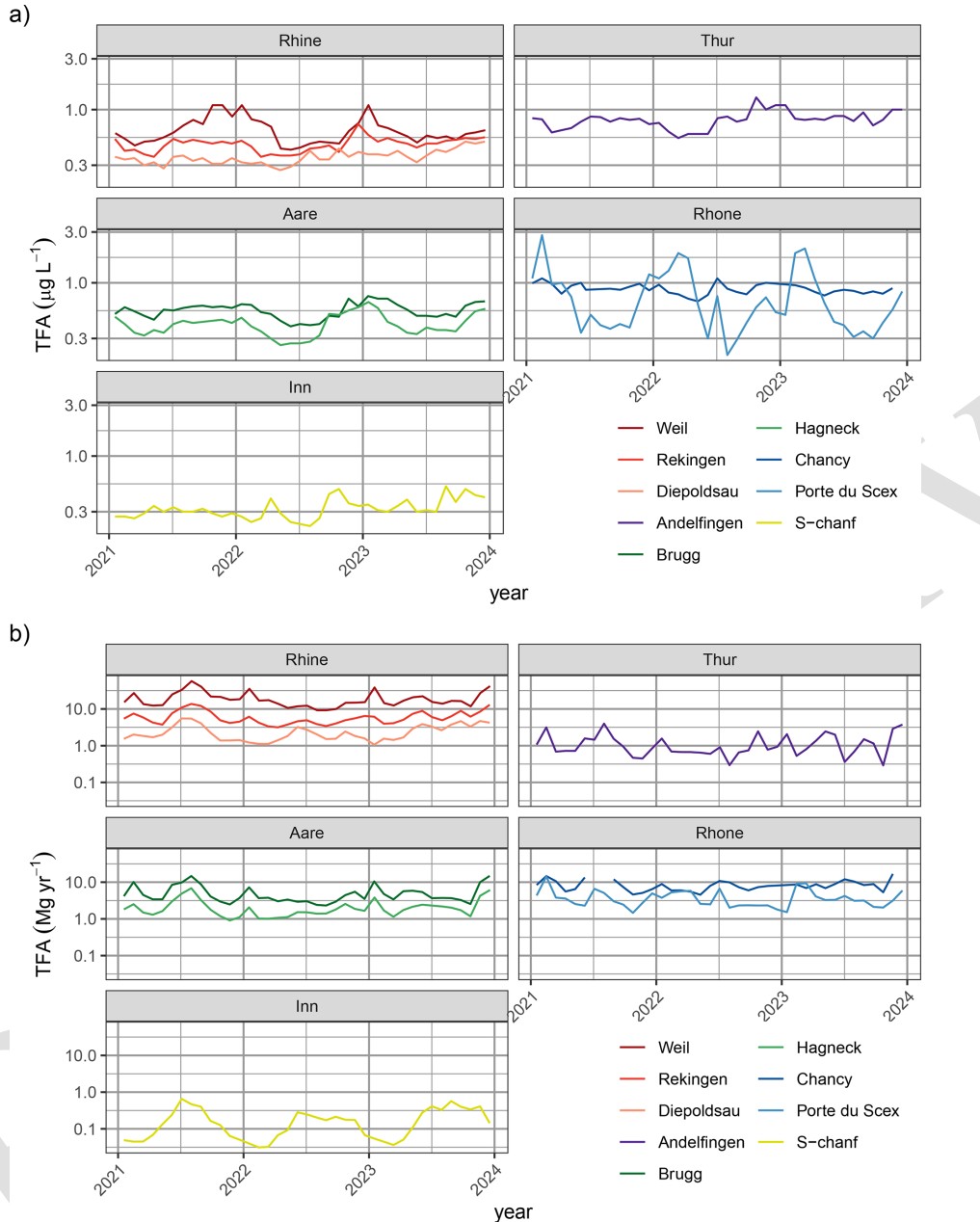

**Figure 3.** Time series of **(a)** measured TFA concentration in monthly river samples and **(b)** estimated TFA loads in rivers. Sites are coloured by different catchments: Rhine (red), Thur (purple), Aare (green), Rhone (blue), Inn (yellow). Darker colours represent sites farther downstream in the respective catchment. Note logarithmic $y$-axis scale.

contain any TFA above the limit of quantification. The sampled glacial ice dates from 1892 to 1978. The indeterminable TFA concentrations in these samples is in line with TFA analysis from Arctic ice samples that consistently indicate very low TFA deposition rates before the end of the 1980s (Pickard et al., 2020; Hartz et al., 2023).

TFA concentrations measured in Swiss lakes were generally in line with those measured in the adjacent rivers (compare Table 1 for average lake concentrations). Concentrations in the Lake Constance (Berlingen, Uttwil) were similar to those measured upriver in the Rhine at Diepoldsau, those in Lake Biel agreed with observations in the Aare at Brugg, and Lake Geneva concentrations were in the same range as those at the Rhone outlet in Chancy. The samples taken at Ghiffa, Lake Maggiore, were taken at the deepest point of the lake (360 m depth) and exhibited the lowest TFA concentrations (0.22–0.28 µg L$^{-1}$), suggesting that increases in surface concentration have not reached the lake bottom yet. However,

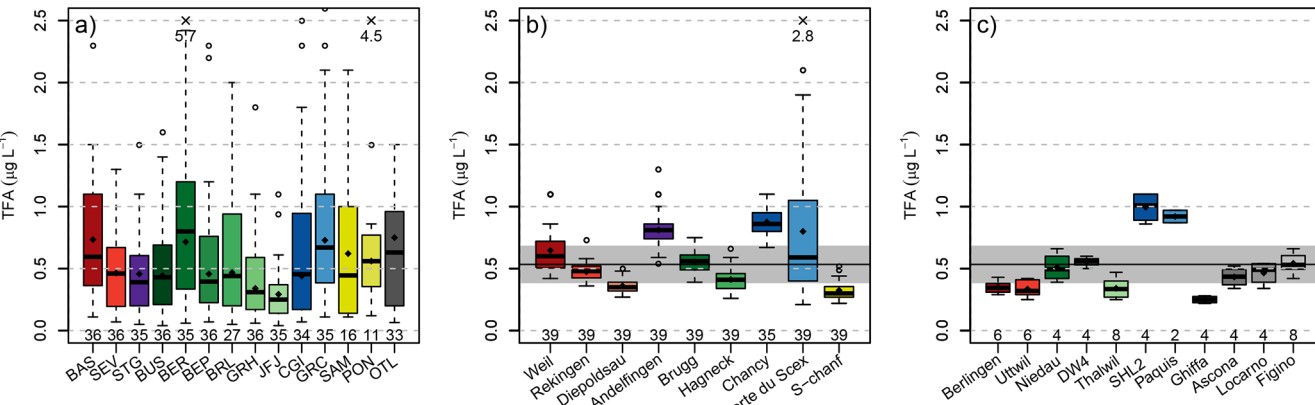

**Figure 4.** Boxplot summary of observed TFA concentrations in **(a)** rainwater, **(b)** rivers, and **(c)** lakes. Boxplot centre lines represent median values, the box the inter-quartile range and the whiskers the range of continuous data (largest/smallest data point that lies 1.5 times the inter-quartile range outside the box). Individual observations outside this range are represented as outliers (empty circles). Numbers below the boxplots give the number of samples at each site. Numbers at the top give values of individual outliers beyond the plot range. The black horizontal line and grey bar in panels **(b)** and **(c)** indicate the average TFA concentration in rainwater. Sites are coloured by different catchments: Rhine (red), Thur (purple), Aare (green), Rhone (blue), Inn (yellow), Ticino (grey). Darker colours represent sites farther downstream in the respective catchment. TS5

higher TFA concentrations (0.34 to 0.54 µg L$^{-1}$ observed in surface water, 20 cm depth, of Lake Maggiore near Locarno and Ascona may have resulted from the nearby agglomeration at the north shore and the river Ticino and related inputs of TFA. No significant TFA differences between lake surface and lake bottom were observed in the Lake Zurich and Lake Geneva (Table 1). Lake water concentrations in the Lake of Geneva were strongly increased compared to all other lakes (1 µg L$^{-1}$), indicating a long term accumulation of TFA, considering mean residence time of lake water of 11.3 years (Commission internationale pour la protection des eaux du Léman, 2024).

## 3.2 Simulated TFA deposition

### 3.2.1 Contribution from atmospheric degradation of long-lived fluorinated compounds

Two different approaches were used to estimate TFA production/deposition from long-lived precursor compounds, a single-box model, which provided a single TFA production rate representative for the years 2021–2023, and a 12-box model, which provided monthly TFA production rates for the period 1995–2023. Both approaches assume that hemispheric TFA production/deposition rates are representative for northern mid latitudes. Applying the single box model, we obtain an average deposition rate of 81 g km$^{-2}$ yr$^{-1}$ for the years 2021–2023. This value is larger than the 62 g km$^{-2}$ yr$^{-1}$ published in Behringer et al. (2021), using the same approach, for the years 2018 due to continued increases in the most relevant HFCs and HCFCs and because a more complete set of compounds was considered here. The by far largest contributor to TFA formation was HFC-134a

with 49 g km$^{-2}$ yr$^{-1}$, which is equivalent to 60 % of the total from all considered HCFCs, HFCs, and FIAs.

The 12-box model simulations (without treatment of HCFC-123, HCFC-133a and FIAs) were initialised in 1995, when, due to the minimal HFC concentrations, TFA contributions from these compounds were close to zero (compare Fig. B1). With the exponential increase of HCFC/HFC concentrations in the following decades the estimated northern hemispheric TFA deposition rate increased likewise and reached an average of 68.3 g km$^{-2}$ yr$^{-1}$ for 2021–2023 (Table 2), which is in very close agreement to the rate estimated from the single-box model, 68.8 g km$^{-2}$ yr$^{-1}$, when only the compounds included in the 12-box model are considered (Fig. B1). Due to the seasonally variable OH concentration and atmospheric temperatures, simulated TFA deposition rates were considerably smaller for winter compared to summer, 40 and 105 g km$^{-2}$ yr$^{-1}$ for the 2021–2023 mean, respectively.

For further comparison with observations, we combine estimates from the single and 12-box model by giving preference to the results from the 12-box model and only employing the constant value from the single box model for HCFC-123, HCFC-133a and the FIAs.

### 3.2.2 Simulated HFO-1234yf and resulting TFA deposition

The detailed simulations of HFO-1234yf degradation can be evaluated both against atmospheric concentration observations of HFO-1234yf itself and against TFA in precipitation. With respect to the first, a reasonable model performance was achieved in terms of temporal correlation, peak magnitude and bias, as demonstrated for the observations at the Swiss

AGAGE site Jungfraujoch (Fig. 5). The model seems to underestimate summertime mole fractions as seen especially in summer 2022 and 2023, which may have to do with an insufficient representation of vertical mixing in Alpine terrain during convective conditions. Model performance for HFO-1234yf was similar for the observations performed during a seven month campaign on the Swiss Plateau, Sottens, in 2021 (Rust et al., 2023). Satisfactory results were also obtained for Monte Cimone and the two AGAGE sites on the British Isles (Mace Head and Tacolneston, see Fig. B2). However, at Mace Head on the Irish West coast and during winter, the simulation underestimated HFO-1234yf levels, which we attribute to HFO transport from North America, not covered by the simulation. Nevertheless, the good agreement of simulated HFO-1234yf in the Swiss domain confirms the general suitability of the model approach.

Simulated daily deposition rates of TFA resulting from the degradation of HFO-1234yf at two representative sites in Switzerland are given in Fig. 6. Deposition rates at the pre-Alpine, sub-urban location Belp were generally larger than at the inner-, high–Alpine site Jungfraujoch. Large deposition rates occurred during the warmer season, whereas winter time deposition was often very small. Individual peak deposition rates equivalent to up to $2.5\,\mathrm{kg\,km^{-2}\,yr^{-1}}$ were forecasted, highlighting the large temporal variability as can be expected from individual precipitation events. This makes the comparison of TFA concentration/deposition observed in earlier studies difficult, as these often sampled individual rain events and not total monthly precipitation.

Average spatial distributions of TFA deposition rates resulting from HFO-1234yf degradation for the period 2021–2023 for Europe and Switzerland are shown in Fig. 7. In general, deposition patterns are not similar to HFO-1234yf emission patterns, which exhibit maxima in the densely populated Benelux region and other European population hotspots (Fig. A2). In contrast, TFA deposition maxima often align with regions of increased precipitation (coastlines, mountain chains) and more intensive photochemistry. Especially pronounced are the high deposition rates in the southern pre-Alps, north of the Po Valley. These can be explained as the result of large HFO emissions in the Po Valley, slow transport in the region due to trapping between Alps and Apennines mountains, relatively fast photochemical processing due to cloud free conditions in the prevailing Mediterranean anticyclone, and finally intense precipitation on the southern flank of the Alps. Next to the dominating deposition maximum south of the Alps, a secondary maximum is present in Switzerland north of the Alps with an emphasis in eastern Switzerland. In contrast, deposition rates are strongly reduced in the inner-Alpine areas, indicating the rapid washout of TFA as simulated by the model.

### 3.2.3 Comparison to observed TFA deposition

When combining the simulations of TFA deposition from HCFCs, HFCs, and FIAs and HFOs (explicitly simulated HFO-1234yf plus approximated contribution from other HFOs), a direct comparison to observations can be attempted. Compared with the observed TFA deposition rates, the simulations captured the seasonal cycle very well, as demonstrated for the sites Jungfraujoch and Belp (Fig. 8). However, summertime deposition was largely underestimated by the simulations. Overall, the simulated deposition rates were able to explain around 60 % of the the observed monthly variability ($R^2 > 0.6$). In the simulations, the seasonal cycle was dominated by the TFA contributions from HFO degradation and reached near zero contributions in winter. In contrast, TFA from HCFCs, HFCs, and FIAs was simulated with a less pronounced seasonality. On the one hand, this is caused by some of the compounds being treated in the single-box approach without seasonal variability of lifetimes. On the other hand, hemispheric loss rates were taken from the 12-box model, which may overestimate the wintertime loss in mid-latitudes. Observed and simulated summertime deposition rates were considerably smaller at the Alpine site Jungfraujoch as compared to the Swiss Plateau site Belp. Although the distance between the sites is only about 50 km, this gradient is also qualitatively captured in the HFO-1234yf simulations.

The large summertime underestimation of TFA deposition rates in the model also leads to an underestimation of average annual deposition rates, which varies with the location of the investigated sites. For sites on the Swiss Plateau 60 %–70 % of the observed TFA deposition rates can be explained by the simulation, with 40 %–54 % of the observed TFA originating from HFO degradation and 12 %–17 % from HCFC, HFC, and FIA degradation (Fig. 9). The explained fraction was considerably smaller for two sites in more urban/suburban environments (Bern, Basel-Binningen), which may indicate additional anthropogenic primary sources of TFA to the atmosphere. Secondary production from very short-lived compounds could play a role as well, whereas compounds considered here have sufficiently long lifetimes, such that local enhancements of TFA close to the emission source is less likely. For sites in mountainous terrain the fraction of TFA deposition explained by the model was generally smaller than for the Swiss Plateau sites (30 %–40 %), with the exception of the site at Jungfraujoch. This may indicate that the simulated minimum of TFA deposition in the inner Alpine area is too pronounced in the model. The latter could be explained by a too rapid washout of TFA when rain is induced at the flanks of the mountains. In addition, deposition rates at the only site south of the Alps (Locarno-Monti) are largely underestimated (42 % explained). The site is located just north of the TFA deposition maximum simulated for the north-western part of the Po Valley (compare Fig. 7). Again, the washout may be slightly too fast in the model and maxi-

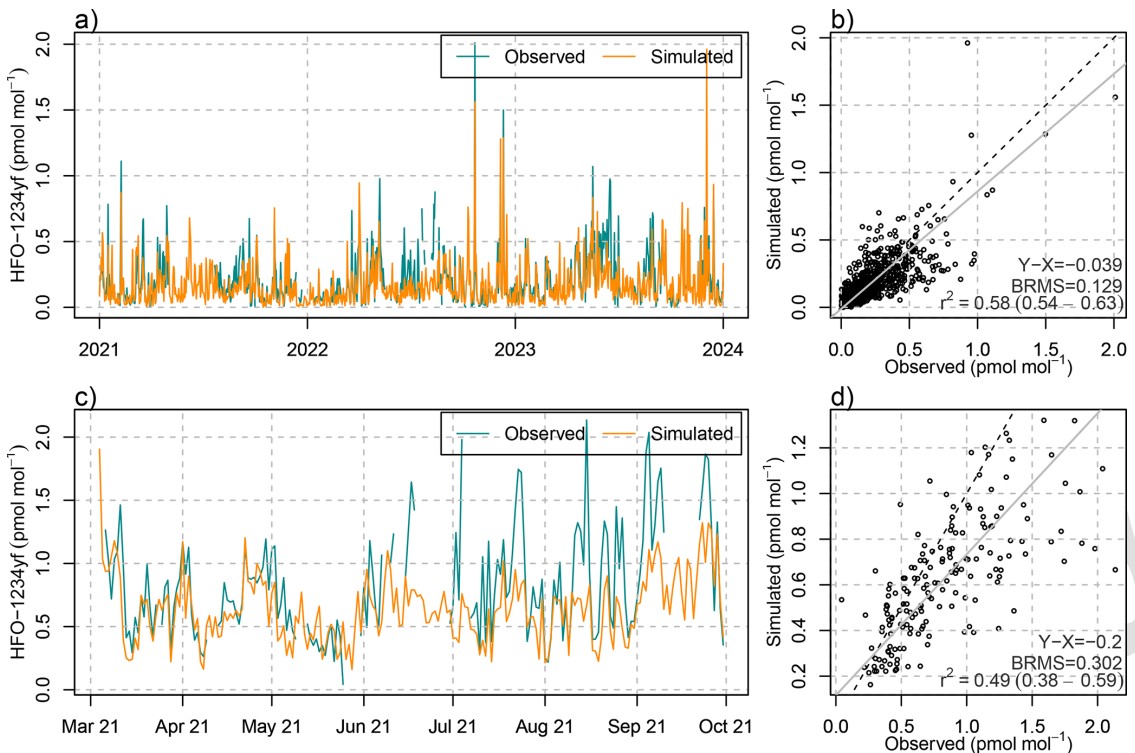

**Figure 5. (a, c)** Temporal evolution of (blue) observed and (orange) simulated daily mean mole fractions of HFO-1234yf at **(a)** Jungfraujoch and **(c)** Sottens. **(b, d)** Scatter plot including (grey) linear regression, (dashed) 1-to-1 line, and comparison statistics: bias ($X$-$Y$), bias-corrected root mean squared error (BRMS), squared Pearson's correlation coefficient, $R^2$, at **(b)** Jungfraujoch and **(d)** Sottens.

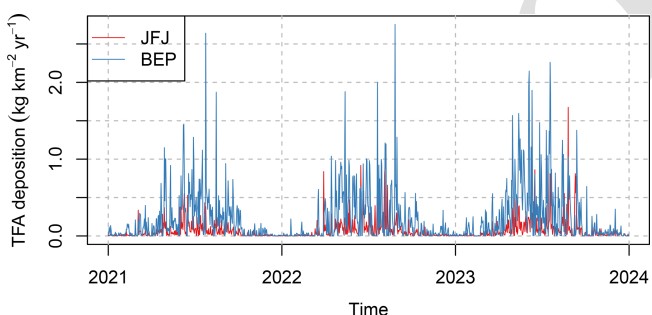

**Figure 6.** Temporal evolution of simulated daily total (wet+dry) TFA deposition at the sites (red) Jungfraujoch and (blue) Belp.

mum deposition may therefore occur too far south. However, such steep gradients in deposition rates remain challenging for the transport model and additional observations south of the Alps would be required for a more thorough characterisation and model validation.

The estimated combined uncertainty of the simulated TFA deposition, accounting for uncertainty in TFA yields and precursor emissions, was about 66 % for the precipitation sites on the Swiss Plateau (compare Fig. C1 and cannot fully explain the unaccounted fraction. Even if upper limit emissions and TFA yields are considered, only about 80 % of observed

TFA can be accounted for by the model for the Swiss Plateau sites (Fig. C1). For the year 2021, simulations with slower and faster OH degradation were carried out. Results in terms of total annual TFA deposition (dry+wet) were compared to those of the base simulation. A 10 % faster OH degradation resulted in an increase of TFA deposition in the Swiss domain of 4 %. Vice-versa, a 10 % slower OH degradation decreased deposition by 4 %. On the European scale differences over the continent remained in a similar order as for the Swiss domain, whereas differences were larger over the more remote maritime areas with clear tendencies to increased and decreased deposition for the slower and faster degradation, respectively. The 10 % percent uncertainty range explored here encompasses more recent kinetic data that suggest a slower best estimate of the OH reaction than used here (Orkin et al., 2010; Tokuhashi et al., 2018). Garavagno et al. (2024) explored the impact of the slower reaction rates by Tokuhashi et al. (2018) on near-surface TFA concentrations in an atmospheric transport and chemistry model and derived between 2 % and 5 % higher concentrations over Europe, which agrees well with our estimate. Therefore, uncertainty on the reaction rate of the initial degradation step does not seem to be a major contributor to unaccounted TFA deposition. An attempt to reconcile the simulated and observed TFA deposition rates is presented in Sect. 3.5.

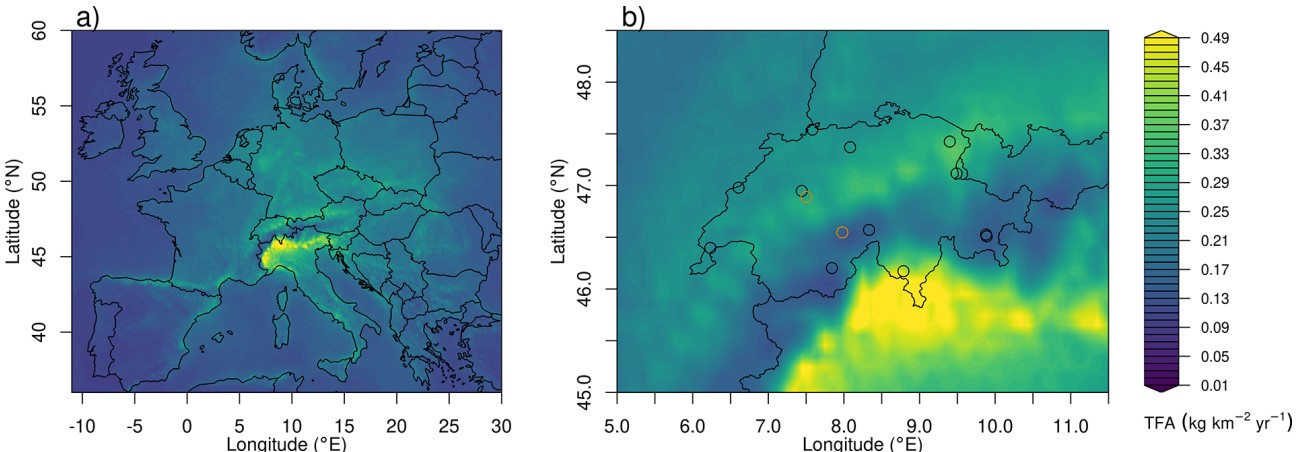

**Figure 7.** Average of total (wet+dry) TFA deposition as derived from HFO-1234yf degradation for the period 2021–2023 **(a)** across Europe and **(b)** within Swiss domain. The symbols mark the locations of the precipitation observations, with orange circles referring to the sites Bern and Jungfraujoch, for which time series of TFA deposition are given in Fig. 6.

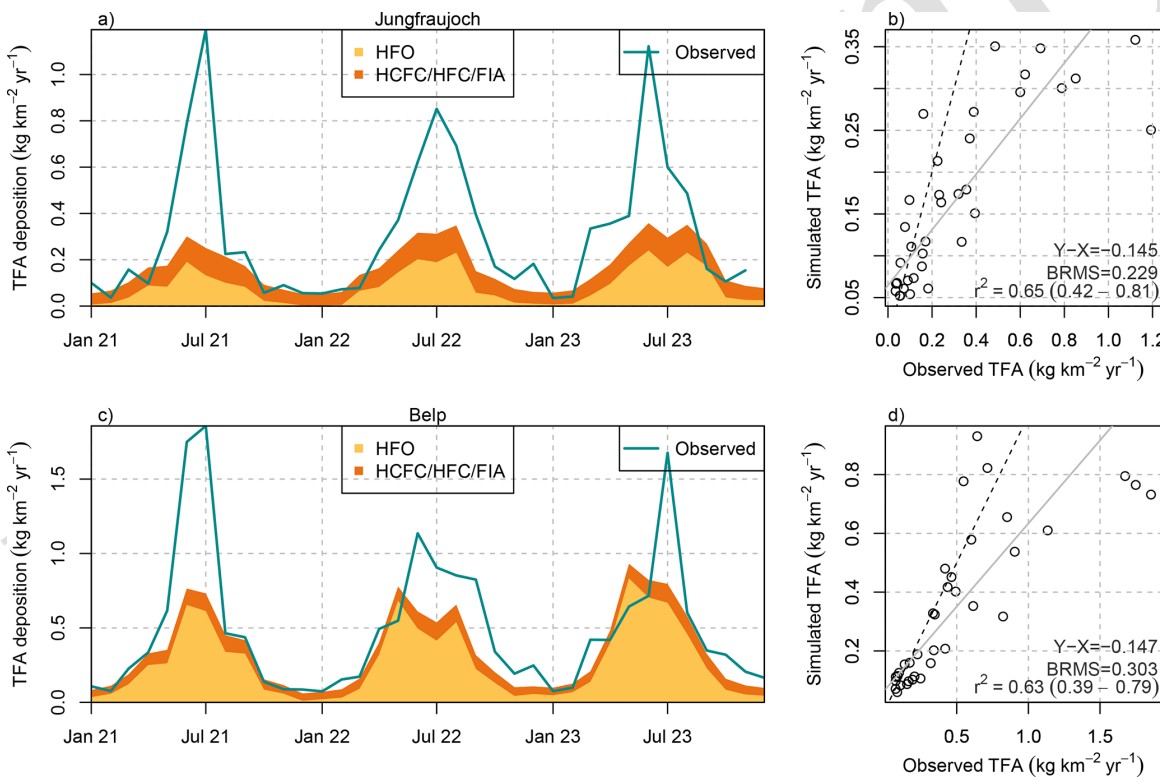

**Figure 8.** Comparison between monthly observed (green line) and simulated total (wet+dry) TFA deposition from HFOs (yellow) and HCFCs/HFCs/FIAs (orange) at **(a, b)** Jungfraujoch and **(c, d)** Belp. **(b, d)** Scatter plot including linear regression line (grey), 1-to-1 line (dashed), and comparison statistics: bias ($X$-$Y$), bias-corrected root mean squared error (BRMS), squared Pearson's correlation coefficient, $R^2$.

## 3.3 Total TFA fluxes, source attribution and budget in Switzerland

Using the average observed deposition rate (all precipitation sites, 2021–2023), total atmospheric input of TFA in Switzerland is estimated to be $24.5 \pm 9.6\,\mathrm{Mg\,yr^{-1}}$ (mean and standard deviation across observational sites). Using the simulated TFA deposition distribution and TFA deposition rates scaled to the observations (see Sect. 3.5), lower total deposition is obtained: $20.7 \pm 5.3\,\mathrm{Mg\,yr^{-1}}$, which is related to a potential underestimation of deposition rates in the Alps

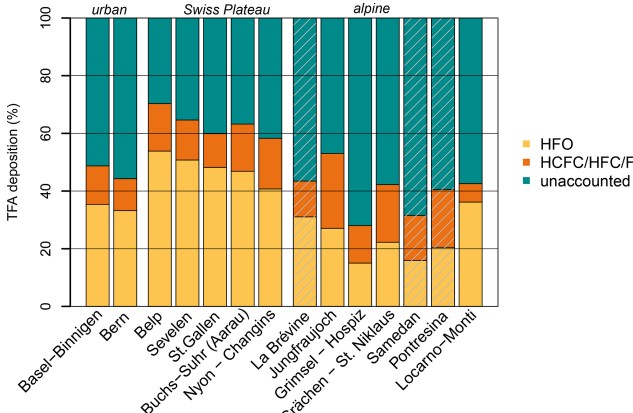

**Figure 9.** Simulated, relative contributions to observed, average TFA deposition at the Swiss monitoring sites for the period 2021 to 2023. Simulated TFA deposition from HFOs (yellow), HCFC-s/HFCs/FIAs (orange), and unaccounted fraction (green). Grey shaded bars indicate sites with an incomplete observation coverage ($< 90\,\%$). Sites Basel-Binnigen and Bern are urban/suburban locations. Sites Belp to Nyon-Changins are located on the Swiss Plateau. Locarno-Monti is located south of the Alps. All other sites are mountain sites, either with the Alps or the Jura mountains (La Brévine).

even after scaling. Of the TFA originating from HFOs, 18 % can be attributed to HFO emissions from Switzerland (CH simulation with only Swiss emissions). For HFO originating from long-lived compounds no explicit attribution to emissions from Switzerland is feasible. However, since hemispheric background mole fractions determine the TFA deposition, the domestic fraction should be similar to the Swiss contribution to global emissions of these compounds, which is smaller 0.3 % for the long-lived compounds considered. In the case of HFC-134a, Swiss emission of $410\,\mathrm{Mg\,yr^{-1}}$ (Swiss Federal Office for the Environment, 2024b) in 2022 can be contrasted by global emissions of $260 \pm 30\,\mathrm{Gg\,yr^{-1}}$ (Western et al., 2025), hence, a Swiss contribution of 0.16 %.

For non-atmospheric sources of TFA, some assumptions can be made for plant protection products (PPP) and veterinary pharmaceuticals. Joerss et al. (2024) estimated for the years 2011 to 2017 the average use of plant protection products, which contain a $CF_3$ terminal group and which may form TFA in the environment. Using their estimate, the annual input of TFA from PPPs to Swiss soils can be calculated as 2.9 to $9.8\,\mathrm{Mg\,yr^{-1}}$. Poiger et al. (2025), taking into consideration additional PPPs and more recent sales numbers from the Swiss Federal Office for Agriculture (2025), estimated an even higher TFA input of up to $11.8\,\mathrm{Mg\,yr^{-1}}$ for the years 2011 to 2023. An additional emission pathway of TFA to soils is the degradation of veterinary drugs, either on pastures after deposition by grazing cattle or on cropland through manure application. Currently, an estimate for these emission routes in Switzerland is $1.2 \pm 0.2\,\mathrm{Mg\,yr^{-1}}$ (Rachel

L. London, personal communication, 25 May 2025), considerably smaller than the above sources.

Altogether, these sources sum up to $33 \pm 11\,\mathrm{Mg\,yr^{-1}}$ TFA. Annual TFA exports from rivers exiting Switzerland were estimated to $31 \pm 4\,\mathrm{Mg\,yr^{-1}}$. This includes the directly measured export at the two major rivers leaving Switzerland (Rhine at Weil and Rhone at Chancy, Table 1). Both sites include additional non-Swiss territories in their catchments. Although, these areas are relatively small compared to the Swiss part of the catchment, the above loads have to be taken as upper limits when applied to the Swiss TFA budget. The flux through the river Inn was only measured relatively high up in the catchment, $0.19 \pm 0.03\,\mathrm{Mg\,yr^{-1}}$, at S-chanf, and a factor of two was applied to this flux to account for the Swiss catchment of the Inn below S-chanf. The flux through the river Ticino in southern Switzerland was not directly monitored. However, from episodical TFA samples taken in September 2024 at Riazzino (Ticino) and Locarno (Maggia), we estimate an approximate flux of $2.6 \pm 1.3\,\mathrm{Mg\,yr^{-1}}$. Similarly, TFA samples for the river Doubs in north-western Switzerland were taken in summer 2023 only, allowing for an approximate estimation of TFA exports by the Doubs of $0.1 \pm 0.06\,\mathrm{Mg\,yr^{-1}}$. Rivers exiting Switzerland without any available TFA observations are the Adige and Adda in south eastern Switzerland. Their catchments and flows are small compared to the major Swiss river basins and, hence, their TFA exports were neglected in the budget.

There may still be considerable imbalance in the presented Swiss TFA budget, depending on the yield of PPP transformation, the assumption of a steady state of the fluxes within the 3 years of TFA observations and unaccounted inputs/sources and exports (e.g., groundwater formation). One previously suspected source is from communal or industrial WWTPs, which discharge into the river system but were not separately sampled in the current study. Thus, representative observations in WWTP discharge would be very valuable to improve the TFA budget and differentiate these from other unknown sources. Given the high mobility and stability of TFA, bigger storage effects seem to be negligible in surface waters within the 3 years of observation. However, lakes with long residence times like Lake Geneva (11 years) form an exception as the stored TFA (approx. 85 Mg, Table 1) corresponds to 10.2 times of the current annual export by the river system.

A local budget can be assessed for the catchment of S'Chanf (Inn) in the Canton of Grison. The average river load at this site was $0.19 \pm 0.03\,\mathrm{Mg\,yr^{-1}}$ (see Table 1), whereas the average deposition rate at the two precipitation sites in the catchment above S'Chanf (Pontresina, Samedan) was $0.48 \pm 0.26\,\mathrm{kg\,km^{-2}\,yr^{-1}}$, which, with a catchment area of $616\,\mathrm{km^2}$, translates into a total TFA input of $0.29 \pm 0.16\,\mathrm{Mg\,yr^{-1}}$. Larger input from precipitation than export through the above ground river, indicates export towards groundwater formation and confirms the expectation of insignificant other sources of TFA in this high Alpine

catchment. Note, that all local WWTP discharge happens below the station at S'Chanf.

To compare the impact of different TFA inputs for different surface land types, we calculated the TFA input per surface area from atmospheric deposition and PPPs, assuming that these occur over the entire Swiss surface and cropland only, respectively (with surface areas taken from Swiss Federal Statistical Office (2025)). Observed TFA fluxes from atmospheric deposition were $0.59 \pm 0.23 \, \mathrm{kg \, km^{-2} \, yr^{-1}}$ (Table 1), whereas fluxes from PPP usage on croplands were $1.7 \pm 0.9 \, \mathrm{kg \, km^{-2} \, yr^{-1}}$. These results indicate that over Swiss cropland PPPs may be the more relevant source of TFA to environmental aqueous phases (i.e., rivers, lakes and groundwater). The additional input of TFA from veterinary drugs via manure application on cropland may further increase TFA input to groundwater below croplands. This is in line with the observation that TFA concentrations in Swiss ground water are highest in areas with large cropland fractions (Swiss Federal Office for the Environment, 2024a). If all degradation of veterinary pharmaceuticals through grazing cattle or manure application is assigned to pasturelands, TFA fluxes similar to those from the atmosphere can be expected, $0.69 \pm 0.14 \, \mathrm{kg \, km^{-2} \, yr^{-1}}$.

### 3.4 Long term trend from archived samples

The assessment of archived precipitation samples for TFA concentrations allows for a long term analysis of TFA deposition trends in Switzerland. The earliest samples analysed date back to the year 1986/1987 and were collected at two sites (Belp and Jungfraujoch) (Fig. 10). Nine of these 25 samples did not contain any TFA above the LOQ of $0.05 \, \mathrm{\mu g \, L^{-1}}$, whereas the other samples showed maximal concentrations of up to $0.23 \, \mathrm{\mu g \, L^{-1}}$ and the precipitation weighted mean (PWM) TFA concentration was $0.085 \, \mathrm{\mu g \, L^{-1}}$. In 1986/1987, only one of the winter samples contained concentrations larger than the LOQ, indicating that atmospheric photochemical production was a main driver to TFA deposition already. Twelve samples taken in 1992, showed a considerably larger PWM concentration of $0.173 \, \mathrm{\mu g \, L^{-1}}$. Because of the small sample size and short period of analysis, it is not possible to relate these elevated observations to a general increase in TFA loads since 1988. Of the eleven samples taken in 1994/1995, only four showed concentrations larger than the LOQ and an overall mean of $0.090 \, \mathrm{\mu g \, L^{-1}}$ (PWM: $0.062 \, \mathrm{\mu g \, L^{-1}}$).

These concentrations can be compared to previously reported TFA concentrations in Switzerland and neighbouring countries. However, earlier reports of TFA concentrations in precipitation often sampled individual rain events and a comparison to monthly mean TFA samples is not straightforward. TFA concentrations in rain samples were analysed by Frank et al. (1995) for samples collected in Bayreuth, Germany, in 1994. Reported concentrations ranged from $< 0.003$ to $0.12 \, \mathrm{\mu g \, L^{-1}}$ for 11 individual rain samples. For March to December 1995, the same authors estimated mean TFA concentrations in 34 rain samples of $0.1 \, \mathrm{\mu g \, L^{-1}}$, with individual samples ranging from $0.03$–$0.24 \, \mathrm{\mu g \, L^{-1}}$, for Bayreuth and concentrations in the range of $0.04$–$0.08 \, \mathrm{\mu g \, L^{-1}}$ for two locations in Switzerland, Zürich and Alpthal (Frank et al., 1996). Extending this time series to the end of 1996, Jordan and Frank (1999) estimated average TFA concentrations in rainwater in Bayreuth of $0.110 \, \mathrm{\mu g \, L^{-1}}$ (range 0.01–$0.41 \, \mathrm{\mu g \, L^{-1}}$). Berg et al. (2000) reported mean TFA concentrations in 73 Swiss rainwater samples from 6 locations collected during 1996/1997 of $0.151 \, \mathrm{\mu g \, L^{-1}}$ (range $< 0.003$–$1.55 \, \mathrm{\mu g \, L^{-1}}$). Overall and given the different nature of the sampling strategy, these earlier analysis are largely in line with our analysis of archived precipitation samples (compare thick crosses in Fig. 10). Compared to archived samples and previously published observations, current 2021–2023 TFA concentrations have increased about four-fold in the last 25 years. This increase in precipitation is similar to trends (two- to five-fold) in archived plant material from Germany between 1989–2020 (Freeling et al., 2022). Similarly, increasing TFA concentrations in wine have been reported within the last decades (Burtscher-Schaden et al., 2025).

Archived precipitation samples from 2000 onwards were analysed more continuously, showing large variability for individual months and a continued increase in concentrations (Fig. 10). For 2011, 36 archived precipitation samples were analysed, yielding a mean concentration of $0.200 \, \mathrm{\mu g \, L^{-1}}$ (PWM: $0.183 \, \mathrm{\mu g \, L^{-1}}$). For the same period and in agreement with the current analysis, Henne et al. (2012a) published PWM concentrations of $0.14$ to $0.24 \, \mathrm{\mu g \, L^{-1}}$ collected at three Swiss sites from April–October 2011. The latter were compiled from 99 weekly total precipitation samples.

In 2018, mean TFA concentrations of $0.379 \, \mathrm{\mu g \, L^{-1}}$ (PWM: $0.307 \, \mathrm{\mu g \, L^{-1}}$) were estimated from 17 archived precipitation samples from Swiss sites. Freeling et al. (2020) analysed 1187 24 h (or multi-day) precipitation samples collected at 8 sites across Germany between February 2018 to January 2019. Precipitation weighted mean TFA concentrations ranged from $0.186$ to $0.520 \, \mathrm{\mu g \, L^{-1}}$ for the different sites, whereas individual 24 h samples reached maximal concentrations of $38.0 \, \mathrm{\mu g \, L^{-1}}$. After 2019, the increase in TFA concentrations in precipitation accelerated reaching mean (all sites) concentrations of $0.643 \, \mathrm{\mu g \, L^{-1}}$ in 2023 (PWM: $0.496 \, \mathrm{\mu g \, L^{-1}}$), which, qualitatively, is in line with a strong increase of HFO emissions in the same period (see Fig. A1).

For the period 1987 to 2011, archived river water samples were analysed for three sites: Weil (Rhine), Porte du Scex (Rhone) and S-Chanf (Inn) (Fig. 11). Different behaviour was observed for the three rivers. TFA concentrations before 2000 showed rather little variability at Weil with concentrations remaining below $0.1 \, \mathrm{\mu g \, L^{-1}}$. Since 2000 concentrations rose exponentially (Fig. 11, note logarithmic $y$-axis). In 2017, Scheurer et al. (2017) reported TFA concentrations in the Rhine in Basel of $0.4 \, \mathrm{\mu g \, L^{-1}}$, in line with the long-term increase and current (2021–2023) levels of $0.65 \, \mathrm{\mu g \, L^{-1}}$ (Ta-

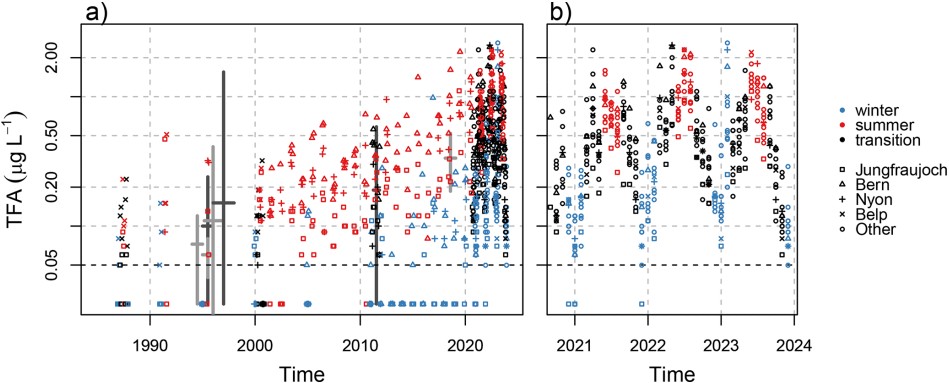

**Figure 10.** Observed monthly mean TFA in rainwater concentrations from **(a)** archived and recent precipitation samples and **(b)** focus on recent monitoring period. Thick pluses represent ranges of historical observations from Switzerland (dark grey) and Germany (light grey). The latter do not represent monthly means but a variety of sampling strategies, see text for details. Values below the limit of quantification (LOQ = 0.05) are plotted at half the LOQ. Note the logarithmic $y$-axis.

ble 1). Further downstream (Scheurer et al., 2017) observed strongly enhanced TFA concentrations in the Rhine (up to $1.3 \, \mu g \, L^{-1}$) and traced these to industrial sources and WWTP discharges. In the Netherlands, TFA is monitored by RIWA at several locations along the Rhine. Annually reported concentrations have largely remained above $1 \, \mu g \, L^{-1}$ since 2017, although peak concentrations (as expected from industrial discharges) have decreased in recent years (Fig. 1.18 in RIWA-Rijn, 2024). Similarly but at a lower level, TFA concentrations increased gradually at S-Chanf from 2000 onwards. For Weil previous observations for 1996/1997 with an average TFA concentration of $0.075 \, \mu g \, L^{-1}$ were reported (Berg et al., 2000), in close agreement with the archived samples. In the upper Bavarian catchment of the river Inn, Christoph (2002) observed 0.04 to $0.08 \, \mu g \, L^{-1}$ TFA in 1998/99. Again, this is in agreement with results from archived samples from S-Chanf taken in 2002.

The interpretation of the archived samples from the Rhone at Porte du Scex, just before it enters Lake Geneva, is less straight forward. Three archived water samples from before 1990 did not contain TFA above the LOQ, whereas after 1990 samples frequently contained large TFA concentrations of up to $4.4 \, \mu g \, L^{-1}$. During the recent sampling period, TFA concentrations once again were more comparable to the other Swiss sites, although some months with high concentration were recorded ($2.8 \, \mu g \, L^{-1}$ in January 2021). The large historic concentrations in the Rhone agree with the large amount of TFA accumulated in the Lake of Geneva and suggest sources other than atmospheric deposition (most likely industrial as outlined in the following). Storck (2017) observed TFA concentrations of 1.7 to $2 \, \mu g \, L^{-1}$ in spring 2017 between Sion and Porte-du-Scex during a longitudinal survey campaign of the Rhone, while concentrations in the upper catchment were much lower (0.2 to $0.32 \, \mu g \, L^{-1}$). Previous elevated TFA inputs are also supported by the findings of Christoph (2002), who detected elevated TFA con-

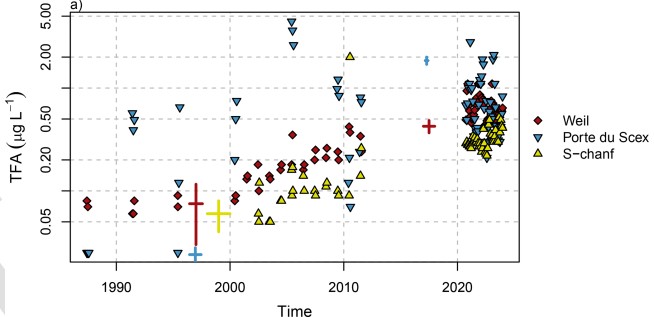

**Figure 11.** Observed TFA concentrations in rivers from archived samples and from recent monitoring period for selected sites. Thick pluses represent ranges of historical observations from the same/close-by locations (Weil and Porte du Scex (Berg et al., 2000) and Basel (Scheurer et al., 2017)) or similar catchment (Inn (Christoph, 2002) and Rhone (Storck, 2017)). Values below the limit of quantification (LOQ = 0.05) are plotted at half the LOQ. Note the logarithmic $y$-axis.

centrations in Swiss spruce needles in the Rhone valley compared to other Swiss sites in 1999 and suspected potential emissions from the regional chemical industry. A potential source could be aluminium smelting using the Hall-Héroult method, which employs cryolite $Na_3AlF_6$ and has the potential to generate fluorocarbons and release these into the environment both the atmosphere and wastewater. Two such aluminium smelters were operational along the Rhone in Chippis and Steg until 1993 and 2006, respectively (Canton Du Valais, 2018).

Compared to average TFA concentrations in Swiss rivers reported at $0.087 \, \mu g \, L^{-1}$ in 1996/1997 (Berg et al., 2000) and in agreement with the archived river water samples, a factor of six increase of TFA concentrations over the last 25 years can be estimated (2021–2023 average: $0.521 \, \mu g \, L^{-1}$, excluding the observations from the Rhone). In 1996/1997

TFA concentrations in precipitation were considerably larger than those in rivers ($0.151\,\mu g\,L^{-1}$ vs. $0.087\,\mu g\,L^{-1}$), whereas for 2021–2023 average concentrations were much more similar in both media ($0.58\,\mu g\,L^{-1}$ vs. $0.521\,\mu g\,L^{-1}$). Together with the fact that TFA concentrations in the uppermost parts of the rivers were generally smaller than those downstream, this could be an additional indicator for increased inputs from industrial waste water and from the degradation of $C-CF_3$-containing PPPs and pharmaceuticals. A very similar six-fold increase in TFA concentrations in stream water was observed in Northern California between 1998 and 2021 (Cahill, 2022). However, they note that 2021 was characterised by little precipitation, such that the comparison was somewhat distorted.

Berg et al. (2000) reported TFA concentrations from "midland" lakes in 1996/1997, these being the more shallow lakes Greifensee, Murtensee and Sempachersee, as compared to the larger and deeper lakes sampled in the current study. The average TFA concentration for these midland lakes was $0.119\,\mu g\,L^{-1}$. Compared to our $0.416\,\mu g\,L^{-1}$ (excluding Lake Geneva) a four-fold increase in the last 25 years is evident in lake water as well. From July 1998 to July 1999, surface water from Bavarian rivers and lakes (14 sites) were sampled and analysed for TFA by Christoph (2002) every 3 month. Average TFA concentrations in rivers were $0.11\,\mu g\,L^{-1}$ (with elevated concentrations of up to $0.22\,\mu g\,L^{-1}$ in the lower Main) and in big lakes a range of $0.05$ to $0.12\,\mu g\,L^{-1}$ were recorded, close to the observations from Switzerland at the time, but excluding the Altmühlsee, which showed TFA concentrations of up to $0.465\,\mu g\,L^{-1}$.

## 3.5 Discussion

### Reconciliation of simulated and observed deposition

TS6 When comparing the long time series of observed TFA deposition rates with those calculated from the degradation of long-lived compounds (symbols vs. thick gray line in Fig. 12), it is obvious that especially summertime deposition rates are largely underestimated by our model assumption for the period 1995 to 2015. Furthermore, we had seen in Sect. 3.2.1 that the combined simulations of TFA deposition from long-lived and short-lived fluorinated gases cannot fully explain observed deposition rates for recent years either. To further quantify and attribute this deficiency we attempt a rescaling of the simulated deposition rates to the observed data.

For the period 1995 to 2015 we assume that TFA deposition is dominated by contributions from long-lived fluorinated gases and we use our estimate for contributions from HCFC, HFC, and FIA degradation as the sole predictor of observed TFA deposition. However, we allow for a seasonally variable scaling factor, such that

$$d_o = d_l + \epsilon = m_1 (a_1 + a_2 \, \sin(2\pi t) + a_3 \, \cos(2\pi t)) + \epsilon, \quad (2)$$

where $d_o$ and $d_l$ represent the observed TFA deposition rates and the one simulated for long-lived compounds, $m_1$ is the original TFA deposition simulated for these compounds, $t$ gives time in years, $a_1$ to $a_3$ are fit parameters and $\epsilon$ is the model error. With this model, we thus estimate the first annual harmonic of a linear scaling factor for our originally simulated values.

Optimising $a_1$ to $a_3$ by employing non-linear least-square regression (Bates and Watts, 1988), we obtain an average scaling factor of 0.83 for wintertime and 4.9 for summertime. The minimum in the scaling factor is estimated for mid January. The scaled HCFCs/HFCs/FIAs contribution is shown by the orange curve in Fig. 12. The scaled model is able to follow the positive trend in TFA deposition rates observed during this period. Hence, we may attribute this trend to growing HCFCs/HFCs/FIAs concentrations. However, for the limited observations before 2000, when concentrations of these compounds were very low, the model cannot explain the larger than zero observations. An alternative regression model that considers a base TFA deposition before the introduction of HCFCs, HFCs, and FIAs simulates peak summertime deposition rates of $0.15\,kg\,km^{-2}\,yr^{-1}$ before 1994. When combined with scaled contributions of HCFCs, HFCs, and FIAs this model did not well describe the increase of deposition between 2000 and 2015. This may indicate that whatever compound contributed to European TFA deposition before 2000, may have declined thereafter. Furthermore and as described above, glacial ice dating back to 1978 did not contain TFA above the LOQ, which would imply an earliest possible onset and slow growth of a significant contribution of an atmospheric precursor to TFA deposition in the Swiss Alps at the beginning of the 1980s.

For the recent period 2021–2023, the scaled contribution from long-lived fluorinated compounds, even when added to the contribution from short-lived compounds, remains insufficient in explaining total observed TFA deposition. Hence, we employ an additional scaling for this period, which we describe by a simple linear factor $b_1$ on the contribution from HFO degradation, $d_s$,

$$d_o = d_l + d_s + \epsilon = d_l + (b_1 m_1) + \epsilon. \quad (3)$$

The fit was only applied to the observations from the Swiss Plateau sites to avoid the potential misrepresentation of inner Alpine sites. The factor $b_1$ was estimated as 1.19, corresponding to an addition of 19 % of TFA from degradation of HFOs or other short-lived precursors. The resulting $d_s + d_l$ is given by the yellow lines (for individual sites) in Fig. 12. Overall, the scaling largely improves the fit to the observations with a near zero bias, a centred root mean-squared error of $0.23\,kg\,km^{-2}\,yr^{-1}$, and a squared correlation coefficient of $R^2 = 0.72$.

For the five-fold underestimation of summertime TFA deposition before the large scale market introduction of short-lived HFOs several reasons may be suggested. Firstly, we

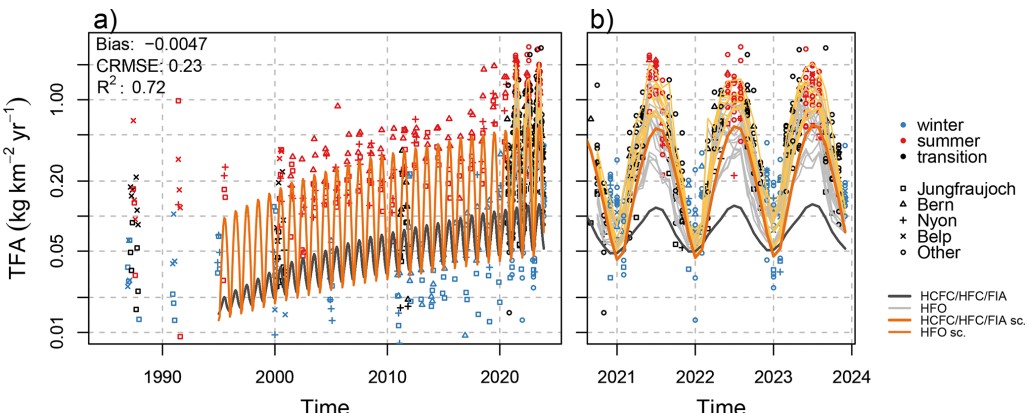

**Figure 12.** Observed monthly mean TFA deposition from **(a)** archived precipitation samples and **(b)** from recent monitoring period. Lines represent different model simulations: (dark gray) TFA from HCFC/HFC/FIA and (light gray) HFO for individual sites in panel **(b)**, (orange) re-scaled TFA from HCFC/HFC/FIA, see text for detail, (yellow) re-scaled TFA from HCFC/HFC/FIA plus re-scaled TFA from HFO-1234yf for individual sites. Note the logarithmic $y$-scale.

may lack information of other TFA-forming precursor substances in the atmosphere. We associated the increase in TFA deposition before 2015 to long-lived compounds. However, the seasonal variability of the derived scaling factor may indicate that some short-lived compounds may be responsible for this process as well. Jordan and Frank (1999) speculated that pyrolysis and thermolysis of fluoropolymers during waste incineration could be a major source of observed TFA, which they were unable to explain by HCFC/HFC degradation. Ellis et al. (2001b, 2003) suggested that the thermolysis of fluoropolymers may occur at temperatures larger than their suggested range of use (e.g., 260°C for PTFE) and be a source of TFA, either through a direct release of TFA at high temperatures or through the formation of hexafluoropropene ($CF_2$=$CFCF_3$, HFP, HFO-1216) followed by atmospheric degradation to TFA with a 100 % yield and an average atmospheric lifetime of 9 d (Mashino et al., 2000). Similarly, (García et al., 2007) detected HFP as the main pyrolysis product of PTFE at 850 °C, whereas HFP yields were small ($< 5\%$) for fuel rich combustion (incineration) of PTFE. Solid waste incineration, as implemented in Europe, is required to perform at high temperatures ($> 850$ °C), which is not likely to produce any TFA or HFP (Wahlström et al., 2021; Améduri, 2023). However, total fluoropolymer waste was estimated to amount to 23.5 Gg in the EU in 2020 (Conversio, 2022), 84 % of which undergo incineration or thermal destruction. A 5 % yield (by mass) of HFP and consequently TFA from fluoropolymer waste treatment would results in 1 Gg TFA and could well explain the larger-than-simulated TFA deposition rates. Another indication for a considerable role of direct or indirect sources of TFA from fluoropolymers could be the increased TFA deposition rates we observed in urban environments (Bern, Basel-Binnigen). Next to the higher population density and, hence, larger potential TFA emissions from direct fluoropolymer thermolysis, for both sites major waste

incinerators are within 5 km of the sampling locations. However, a small-scale waste incinerator is also located close to the site Buchs-Suhr, without observed increases in TFA deposition rates. Dedicated atmospheric observations of TFA and/or HFP would help to improve the quantification of the impact of fluoropolymer waste treatment on TFA deposition.

Secondly, although we evaluated uncertainties in TFA deposition arising from reported uncertainties in TFA yields and were not able to explain the gap in observed deposition solely by using upper limit TFA yields, it may still be possible that yields remain underestimated in the literature and/or that laboratory derived yields are not representative for atmospheric conditions. Potentially, yields are not constant in time and space, but may depend on atmospheric conditions that may favour different degradation pathways. Especially, the TFA yield of the main contributing long-lived fluorinated compounds (HFC-134a) should be re-evaluated, since its currently accepted value of 7 %–20 % is based on a small number of experiments conducted three decades ago (Wallington et al., 1996). With the above scaling factors an annual average yield of 42 % from HFC-134a (summer: 68 %; winter: 16 %) could explain most of observed gap and would be close to yields reported in earlier global chemistry simulations (37 % Kotamarthi et al., 1998). Furthermore, the relative importance of the degradation pathways of the intermediate $CF_3CHO$ (trifluoroacetaldehyde), formed during degradation of some HFCs (e.g., HFC-143a, 236fa, 245fa, 365mfc) and HFOs (e.g., HFO-1234ze(E), HFO-1336mzz(Z), HCFO-1233zd(E)) may differ from what was assumed in the yields used here (see Tables 2 and 3). Three main degradation paths exist for $CF_3CHO$: reaction with OH, photolysis, and hydrolysis. TFA yields from the OH reaction are small, whereas hydrolysis leads mostly to TFA and no TFA is formed in the photolysis path. However, the latter gained recent attention due to the potential of forming HFC-23 (a very stable

compound with a large GWP). Hence, lifetimes of $CF_3CHO$ lifetimes were recently re-evaluated (Baumann et al., 2025; Sulbaek Andersen et al., 2023; Nielsen et al., 2025). The updated results suggest that previously used TFA yields may be too small and that yields will strongly vary across the troposphere. Even with larger yields the degradation of HFCs along this path may not add considerable amounts to TFA deposition (compare Table 2). However, increased TFA yields from the degradation of HFOs (other than 1234yf) could likely explain gaps between observed and simulated TFA deposition rates. Renewed efforts to improve our understanding of degradation routes for fluorinated gases are required to better predict future burdens.

Thirdly, the environmental degradation of longer-chained fluorinated compounds to TFA may partly occur through the atmosphere as well. PPPs or their intermediate degradation products may enter the atmosphere during application or through windblown dust. Transport and dispersion of these compounds as well as further atmospheric degradation may contribute to TFA deposition away from the source. The current study does not provide evidence for such a contribution, since TFA deposition in rural areas were not enhanced compared to urban areas. However, further investigation of the atmospheric fate of PPPs is required. Fluorotelomers are a group of PFAS that are used in a wide range of applications ranging from fire-fighting to food packaging. Thackray et al. (2020) simulated the atmospheric degradation of such compounds in a global chemistry and transport model. Their simulations underestimated observed TFA deposition in mid-latitudes and the Arctic (Pickard et al., 2020) by more than one order of magnitude, suggesting that, given our current knowledge of fluorotelomer chemistry, their atmospheric degradation is not the dominating source of TFA.

Finally, we cannot rule out that underestimated TFA deposition could be caused by the assumption of homogeneity in TFA formation and deposition that was applied in the simple model approach for long-lived fluorinated compounds. On the one hand, we may expect increased deposition rates in regions with increased precipitation as seen for HFO-1234yf degradation (compare Fig. 7). Hence, our box model may miss this increase and therefore overall tend to an underestimation in a precipitation-rich area like Switzerland. On the other hand, similar underestimation of observed TFA deposition was derived from a similar simulation for sites across Germany as well, that experienced different total precipitation amounts (Behringer et al., 2021). Furthermore, spatially resolved atmospheric transport and chemistry simulations of HCFCs and HFC-134a revealed strong latitudinal gradients with maxima in the tropics for resulting TFA deposition, whereas longitudinal gradients were smaller (Kanakidou et al., 1995; Kotamarthi et al., 1998; Holland et al., 2021; Garavagno et al., 2024). Kanakidou et al. (1995) simulated TFA deposition from HFC-134a, HCFC-123, HCFC-124 degradation in Central Europe for the year 2020, assuming a 2.5 times larger TFA yield from HFC-

134a but with forecasted atmospheric concentrations close to those actually observed. Their estimate was smaller than our estimates ($0.034\,\mathrm{kg\,km^{-2}\,yr^{-1}}$ vs. $0.060\,\mathrm{kg\,km^{-2}\,yr^{-1}}$, compare Table 2). Model predictions by Kotamarthi et al. (1998) for TFA deposition from the same species amounted to $0.061\,\mathrm{kg\,km^{-2}\,yr^{-1}}$ (corrected for larger assumed TFA yield) for Central Europe in 2010. Additional detailed and spatially resolved atmospheric chemistry and transport simulations of long-lived fluorinated compounds with explicit description of their degradation chemistry as conducted by Holland et al. (2021) for HFC-134a would help improve TFA deposition estimates from long-lived compounds. The underestimation of TFA production/deposition from HFOs, estimated to about 19 % of the current model estimate, may well be explained by uncertainties in the TFA yields and emission strength/distributions of these compounds and the neglected non-European HFO emissions. Development of more realistic, spatially resolved emission inventories would allow for more detailed chemistry simulations and validation against observations in the future.

## 4   Conclusions

TFA concentrations are increasing in the environment due to continued use of fluorinated precursor substances such as fluorinated gases, plant protection products, pharmaceuticals and a switch to short-lived, high-yield compounds (HFOs compared to HCFCs and HFCs). For the first time, the current study presents a comprehensive observation and modelling study quantifying and attributing TFA sources, burdens and exports for a country as a whole. At the same time current measurements are put in a historical perspective.

TFA concentrations in precipitation have increased in Switzerland by at least a factor of four in the last three decades, reaching average levels of $0.58\,\mathrm{\mu g\,L^{-1}}$ in 2021–2023. Similarly, TFA concentrations in the major river network have increased six-fold with average concentrations of $0.52\,\mathrm{\mu g\,L^{-1}}$ in 2021–2023 (excluding the industrially highly impacted river Rhone with an average of $0.8\,\mathrm{\mu g\,L^{-1}}$). Average atmospheric TFA deposition rates were $0.59\,\mathrm{kg\,km^{-2}\,yr^{-1}}$. A large fraction of the observed deposition was directly associated by means of atmospheric modelling to the degradation of long-lived HCFCs/HFCs/FIAs (12 %–17 %) and short-lived HFOs (40 %–54 %). The largest contributing long-lived precursor was HFC-134a, whereas HFO-1234yf constitutes the single most important short-lived precursor, with currently quickly growing emissions in Europe. The remainder of the observed atmospheric deposition (29 %–48 %) was not explained by the simulated degradation of known precursor compounds. This gap may be associated with unknown precursor compounds, underestimated TFA yields from known precursors, model shortcomings or a combination of these. Renewed scientific efforts are required to improve our understanding of atmospheric chem-

istry and fate of fluorinated compounds in the atmosphere as well as in soils and water bodies.

In the future, HFO usage in Europe is expected to increase further and may reach 23 to 47 Gg yr$^{-1}$ by the year 2050 (Behringer et al., 2021), whereas atmospheric concentrations of HFC-134a may not have decreased significantly by then (Velders et al., 2022). Assuming a linear concentration/deposition response to emissions, expected TFA deposition rates for Switzerland and other parts of Central Europe may then reach levels 10–20 times larger than now (5.8–11.6 µg L$^{-1}$), close to or above recommend safety thresholds for drinking water (Adlunger et al., 2021; RIVM, 2025). Common water treatment technologies for potable water like activated carbon or ozonation are not suitable to remove TFA, while ion exchange and reverse osmosis can be used (Scheurer et al., 2017). However, economic and ecological criteria often prevent large scale application of the latter both techniques.

Therefore, it is fundamental to continue efforts to abandon all non-essential uses of TFA and its precursors, to avoid further, continued accumulation of TFA. Both industry and policy makers are called to increase their level of ambition. Also, continued continent-wide monitoring, including human biomonitoring, will be necessary to surveil progress and to further improve both our understanding of TFA budgets and our ability to forecast future burdens by atmospheric simulations.

## Appendix A:  European HFO-1234yf emissions

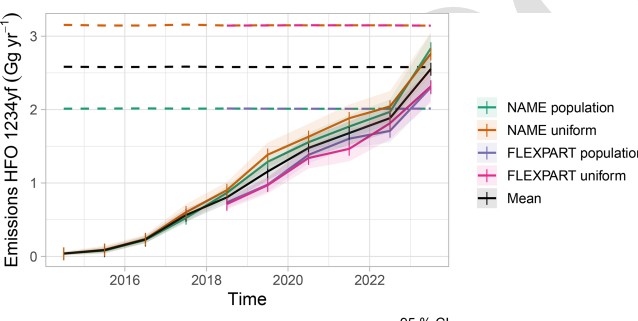

**Figure A1.** Total annual emissions of HFO-1234yf as estimated by atmospheric inverse modelling for EU27 plus Norway, Switzerland, United Kingdom. Results are taken from Vollmer et al. (2025) and represent four sensitivity inversions carried out with Empa's Lagrangian regional inversion system (ELRIS) using two different a priori distributions (population-based and uniform land) and two different atmospheric transport models (NAME and FLEXPART). Dashed lines represent a priori estimates, solid lines a posteriori results, and the black line the mean over the four sensitivity inversions, which was used as input for the HFO-1234yf degradation simulations.

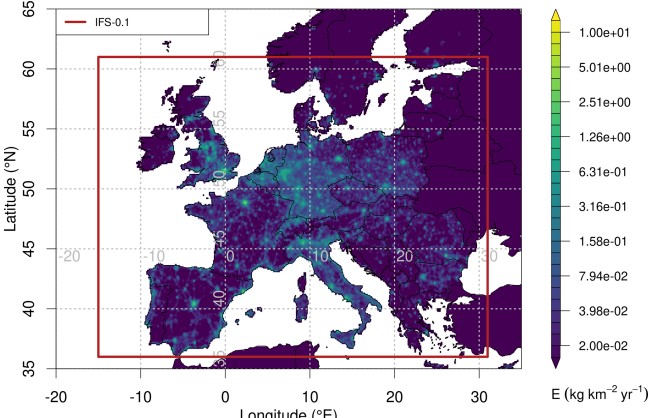

**Figure A2.** Annual mean emissions of HFO-1234yf as used in FLEXPART simulations for the year 2023. The red box marks the domain in which FLEXPART was driven by high resolution ECMWF input at $0.1° \times 0.1°$. Note that emissions outside the EU27+ countries were set to zero.

## Appendix B:  Additional simulation results

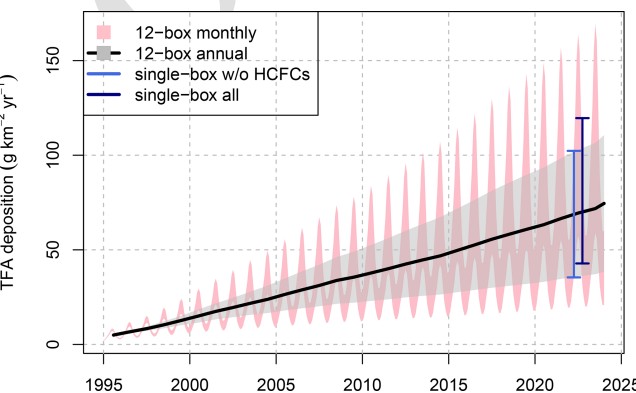

**Figure B1.** Temporal evolution of simulated mean hemispheric TFA deposition rate from HCFC/HFC/FIA degradation: (pink ribbon) monthly and (gray ribbon and black line) annual rates of all considered HFCs and HFCF-124 using the 12-box model, (light blue error bar) 2021–2023 mean from single-box model for HFCs and HCFC-124, (blue error bar) 2021–2023 mean from single-box model for all considered HFCs/HCFCs/FIAs. The uncertainty estimate is based on the range of TFA yields given in Table 2 assuming fully correlated uncertainty between compounds (see Sect. C).

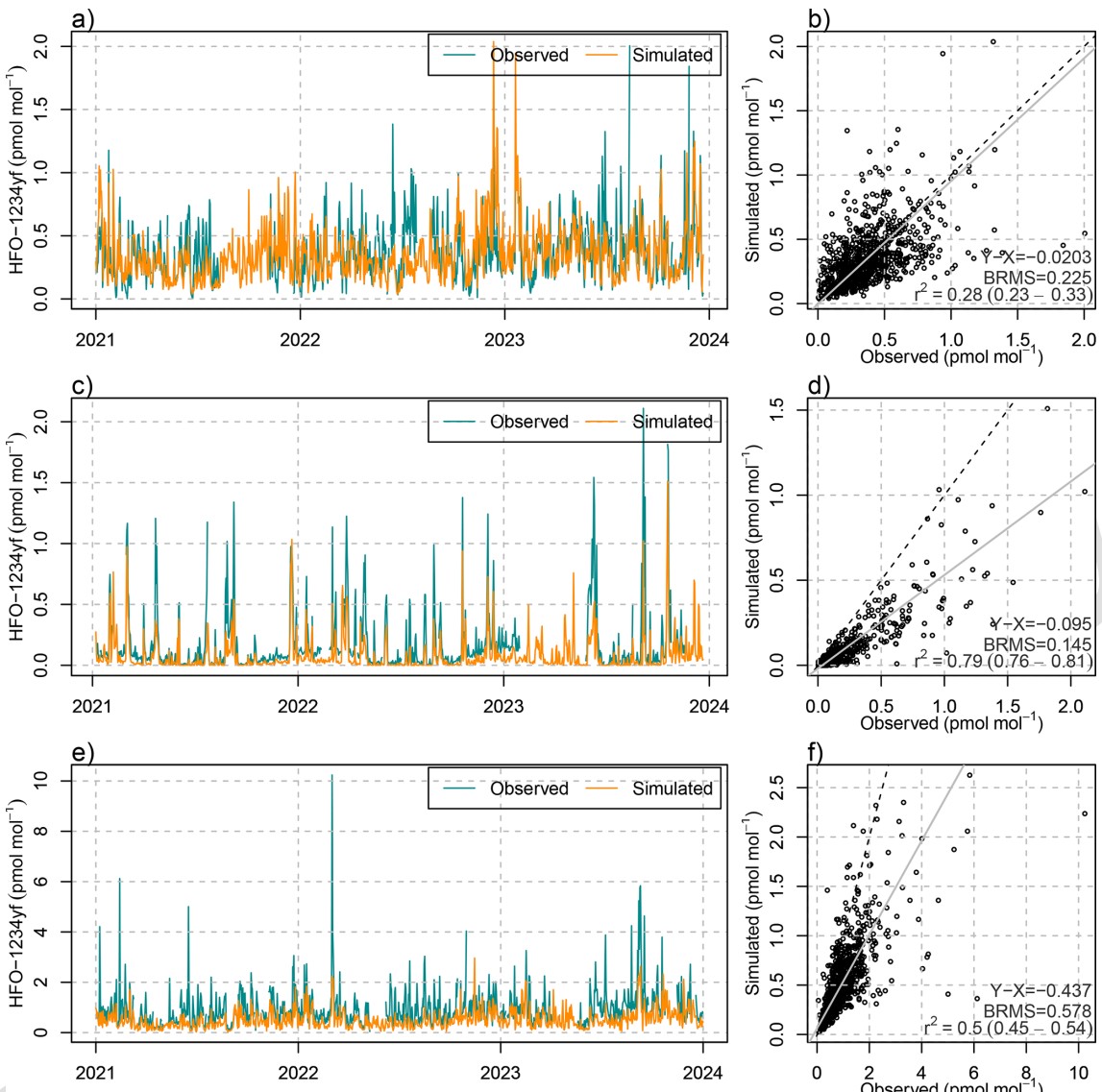

**Figure B2.** **(a, c)** Temporal evolution of observed (blue) and simulated (orange) daily mean mole fractions of HFO-1234yf at **(a)** Monte Cimone, **(c)** Mace Head, **(e)** Tacolneston. **(b, d, f)** Corresponding scatter plots including linear regression (grey), 1-to-1 (dashed) line, and comparison statistics: bias, bias-corrected root mean squared error, squared Pearson's correlation coefficient.

## Appendix C: Uncertainty estimate of simulated TFA deposition rates

The uncertainty of simulated TFA deposition rates for both long-lived and short-lived compounds was assessed as follows. For long-lived compounds we consider two sources of quantifiable uncertainty: simulated loss rates and TFA yields. The uncertainty of the former is taken from the a posteriori uncertainty estimate of global emissions for individual compound as estimated with the 12-box model. This estimate contains the uncertainty of the atmospheric observations propagated through the inverse modelling step and an additional lifetime uncertainty (see Rigby et al., 2008). For most compounds the by far larger uncertainty originates from the TFA yields. We assume that the values given in Table 2 represent the 95 % confidence range of the yields. We use Gaussian error propagation to combine both sources of uncertainty and sum over all long-lived compounds, where we assume fully uncorrelated uncertainty between yields and emissions and fully correlated uncertainties for the yields between compounds, since many of these come from uncertainties of yields from intermediate compounds. Similarly, we combine the uncertainty estimate on European emissions and yields for the additional HFO/HCFOs (Table 3). Not formally quantified in this calculation are other sources of model uncertainty (transport, reaction and deposition rates) and representativeness, which are discussed qualitatively in the main text. The resulting uncertainty range for the relative contributions to observed deposition rates (Fig. C1) represents the 95 % confidence range of this assessment.

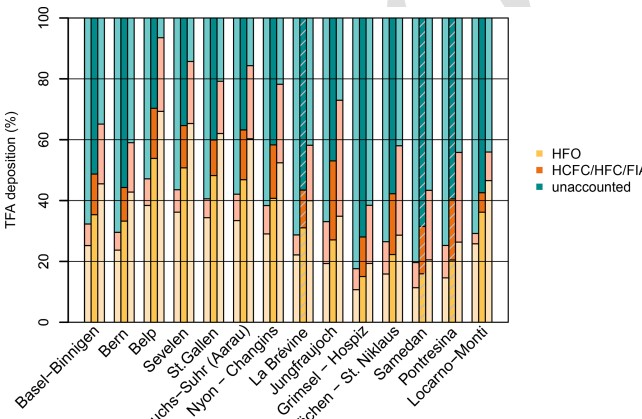

**Figure C1.** Simulated, relative contributions to observed, average TFA deposition at the Swiss monitoring sites for the period 2021 to 2023. Simulated TFA deposition from HFOs (yellow), HCFCs/HFCs/FIAs (orange), and unaccounted fraction (green). Lighter shaded bars indicate the uncertainty range estimated for the simulations. Grey shaded bars indicate sites with an incomplete observation coverage (< 90 %).

**Code and data availability.** The observational TFA data, the FLEXPART-simulated time series of TFA atmospheric deposition at the discussed sites, FLEXPART monthly TFA deposition maps, as well as simulated TFA deposition rates from long-lived compounds are publicly available from the following DOI: https://doi.org/10.5281/zenodo.17220023 TS7 (Henne et al., 2025). HFO observations and emission estimates are part of Vollmer et al. (2025) and can be obtained there. The FLEXPART model code used in this study is publicly available on gitlab.com (https://gitlab.com/empa503/atmospheric-modelling/flexpart_ifs_empa.git TS8, branch TFA, login required). TS9

**Author contributions.** The concept of the study was designed by FRS, HW, MKV, SR, and SH. FRS supervised and carried out precipitation and surface water sampling. MKV conducted halocarbon observations at Jungfraujoch. ML provided archived precipitation samples and access to ice core material. SH carried out atmospheric simulations, prepared and wrote the manuscript with inputs and revisions from all co-authors.

**Competing interests.** The contact author has declared that none of the authors has any competing interests.

**Acknowledgements.** FLEXPART simulations were carried out at the Swiss National Supercomputing Centre (CSCS) under institutional contract ID em05. Precipitation data have been provided by the Swiss Federal Office of Meteorology and Climatology, MeteoSwiss. Two sensitivity runs of the inverse modelling of HFO-1234yf emissions were based on transport simulations using the NAME model and were carried out by Alistair Manning (UK MetOffice). Lucas Passera and Markus Gerber (FOEN) managed the database of TFA observations and provided support in data collocation and analysis. Marc Schürch (FOEN) managed the ISOT network and provided access to precipitation samples. Loïc Schmidely (FOEN) contributed with comments to the manuscript. Laboratories AUA of Eawag and RÜS/AUE BS supported sample handling and shipping. Discharge data for load calculations was provided by FOEN. The measurements of HFOs, HFCs and IFAs at Jungfraujoch are funded by the Swiss National Programs HAL-CLIM and CLIMGAS-CH (Swiss Federal Office for the Environment, FOEN), and supported by the International Foundation for High Altitude Research Stations Jungfraujoch and Gornergrat (HF-SJG). The measurements at Monte Cimone are supported by the Department of Pure and Applied Sciences of the University of Urbino and by the Italian component of ACTRIS (Aerosol, Clouds and Trace Gases Research Infrastructure), under the Programma Operativo Nazionale Ricerca e Innovazione 2014–2020 PIR01 00015

"PER-ACTRIS-IT". Operations at Mace Head and Tacolneston are supported by the University of Bristol through contracts from The Department for Energy Security & Net Zero (prj_1604) and NASA (80NSSC21K1369). Last but not least the authors would like to thank anonymous reviewers and T. J. Wallington, S. Madronich, O. J. Nielsen, K. R. Solomon, M. P. Sulbaek Andersen for their constructive comments that helped improve this manuscript.

**Financial support.** This research has been supported by the NAME OF FUNDER (grant no. GRANT AGREEMENT NO). TS10

**Review statement.** This paper was edited by Ivan Kourtchev and reviewed by three anonymous referees.

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

## Remarks from the typesetter