# Peer review of "Trifluoroacetate (TFA) in Precipitation and Surface Waters in Switzerland: Trends, Source Attribution, and Budget"

_EGUsphere, 2025_

## Author Comment (AC1)

**Reply to 'Comment on egusphere-2025-2861' by Anonymous Referee #1**

Comments are copied in plain text.

Replies are given in blue.

*Changes to manuscript are highlighted in green.*

The authors discuss the trends, budget and deposition of TFA in Switzerland. The study is very comprehensive, including measurements of TFA in rainwater, rivers, lakes, modelling of the deposition of TFA from fluorinated gases, and estimated of TFA from pharmaceuticals and plant protection products. The paper has a good and extensive introduction of TFA. The methods, measurements and model calculations are described in great detail. The results are also described in detail with all their uncertainties and caveats. I want to compliment the authors with this paper, well done. I only have some smaller textual comments.

We would like to thank the referee for the positive response and compliments.

Some specifics comments on the text:

L9: Specify the region the deposition of 24.5 Mg applies to.

These totals were calculated for Switzerland. This was clarified in the revised manuscript.

L11-12: "In croplands … deposition". I suggest removing this sentence. Although interesting, it disrupts the flow of the abstract and is does not add relevant information for the remainder of the abstract.

We agree that the location for this information was not chosen carefully. However, we think that it is an important result especially in the light of the ongoing discussion of TFA in ground and drinking water. Hence, we somewhat rephrased the sentence and attached it to the previous statement concerning total TFA inputs for PPP and veterinary pharmaceuticals.

*"In Switzerland, atmospheric (wet+dry) deposition amounted to 24.5+/- 9.6 Mg yr$^{-1}$, whereas TFA terrestrial inputs from the degradation of plant protection products (PPP) and veterinary pharmaceuticals in soils, estimated from the literature, ranged from 3.9 to 13.2 Mg yr$^{-1}$, depending on the assumption on degradation efficiency. TFA inputs from the degradation of PPP dominated 2-3 times over atmospheric deposition in Swiss croplands."*

L12: It is not clear how old the "archived samples" are. Are they from the 1990s?

This information was given in section 2.1 but not repeated in the abstract. For clarification, we now added it to the abstract as well.

*"Archived precipitation samples from the period 1986 to 2020 revealed that TFA ..."*

L17: You write about the risk assessment, but later in the paper you mention that there is no consensus of the risks of TFA (L81). I suggest you rephrase the sentence, e.g. "for refining the assessment of TFA sources for potential health and environmental risks.".

We would like to thank the referee for this suggestion and have largely adopted it.

*"Additional environmental monitoring and source attribution studies are paramount for refining the assessment of TFA sources and levels for potential health and environmental risks"*

L53-58: This a very long sentence and therefore hard to read. Please rephrase.

We agree that this sentence got a bit out of hand. We broke it up into two parts.

*Due to their long atmospheric lifetime (order of years) and negative impact on global warming and/or the depletion of the ozone layer, national (e.g., EU Regulation on fluorinated greenhouse gases 2024/573 (European Parliament and Council, 2024), American Innovation & Manufacturing (AIM) Act of 2020 (Office of the Law Revision Counsel of the United States House of Representatives, 2020)) and international (Montreal Protocol) regulations are starting to target the emissive use of HCFCs and HFCs. Hence, a transition towards the use of HFOs with short atmospheric lifetimes (order of days to weeks) is ongoing.*

L474: Data from WWTP could indeed be valuable, but are they a new source not accounted for yet? Do the WWTPs not discharge their water on the rivers?

Yes, WWTPs discharge into rivers and as such any TFA originating from them would be accounted for in the downstream observations. What we wanted to express is that right now, we are unable to distinguish this contribution to the total river flux exiting the country. Although inputs from atmospheric deposition and PPP degradation are mostly balanced by the outflow through the river systems, there may be additional sources (like WWTPs) and sinks to ground water formation that we did not quantify. Direct samples at individual WWTPs would help to further close the budget. We somewhat rephrased the paragraph to clarify.

*There may still be considerable imbalance in the presented Swiss TFA budget, depending on the yield of PPP transformation, the assumption of a steady state of the fluxes within the 3 years of TFA observations and unaccounted inputs/sources and exports (e.g., groundwater formation). One previously suspected source is from communal or industrial WWTPs, which discharge into the river system but were not separately sampled in the current study. Thus, representative observations in WWTP discharge would be very valuable to improve the TFA budget and differentiate these from other unknown sources.*

L657: "The gap in explained deposition…". Where the gap refers to is not clear from the previous sentences. Please add some text here.

This comment refers to L677 not L657. For clarification we added one more sentence defining which gap we were referring to.

*The remainder of the observed atmospheric deposition (29-48 %) was not explained by the simulated degradation of known precursor compounds. This gap may be associated with unknown precursor compounds, underestimated TFA yields from known precursors, model shortcomings or a combination of these.*

P30, table 2: There is a typo in the compound name: It should be HFC-43-10mee, not HFC-41-10mee

We would like to thank the reviewer for spotting this. It was corrected.

---

## Author Comment (AC2)

**Reply to 'Comment on egusphere-2025-2861' by Anonymous Referee #2**

Comments are copied in plain text.

Replies are given in blue.

*Changes to manuscript are highlighted in green.*

The authors present a comprehensive assessment of TFA concentrations recorded in precipitation and surface waters in Switzerland 2021-2023. They compare these measurements with those taken from archived water samples and demonstrate significant increases in TFA concentrations in recent years. They also conduct a number of modelling studies to determine TFA deposition from the atmosphere and determine that this cannot be the sole driver of the observed TFA increases. They propose terrestrial inputs from PPP and veterinary pharmaceuticals as a more significant contributor to TFA contamination in surface waters. They also suggest an atmospheric TFA source that predates the introduction of known precursors (though it seems they are suggesting a further anthropogenic source, in which case this should be made clearer in the abstract).

The report is generally well written and presents an important study into TFA contamination at a national level. However, there a few areas that need further discussion or that have been omitted.

Firstly, there should be full justification of the veracity of the archived water samples as these form a major basis for many of the conclusions drawn.

Archived water samples were analysed for stable water isotopes signature and stored accordingly to prevent contamination from the atmosphere during storage (amber glass bottles with screw lock and rubber septa, in the dark, at 12-14°C room temperature). Repeated isotope analysis after more than 20 years of storage revealed no signs of contamination in the water isotopes. Contamination by other compounds is without compromising the water isotopic signature is extremely unlikely. We rephrased the paragraph as follows:

*"Archived samples were stored in amber glass bottles capped with screw tops and inserted septa at temperatures of 12 to 14 °C (Schotterer et al., 2010) under predominantly dark conditions in a cellar. Storage conditions were checked for long-term stability of stable water isotope consistency by redetermination of stable isotopes using Cavity Ring Down Spectrometry after decades of storage (Leuenberger and Ranjan, 2021) in comparison to previous conventional determination by mass spectrometry on these samples right after sampling as part of the IAEA GNIP programme (International Atomic Energy Agency, 2025)."*

Secondly, the relationship between CF$_3$CHO and TFA formation has been omitted from this study. Given the recent literature on the potential impacts of CF$_3$CHO on TFA concentrations and the uncertainty it introduces to TFA yields etc. it should at least be discussed and could go some way to closing the gap seen between the modelled deposition and the measurements.

We would like to thank the referee for this comment. We have included a brief discussion of the degradation of CF3CHO and how recent studies may change the best estimate of TFA yields in our discussion of the gap between observed and simulated deposition rates.

*Furthermore, the relative importance of the degradation pathways of the intermediate CF3CHO (trifluoroacetaldehyde), formed during degradation of some HFCs (e.g., HFC-143a, 236fa, 245fa, 365mfc) and HFOs (e.g.,*

*HFO-1234ze(E), HFO-1336mzz(Z), HCFO-1233zd(E)) may differ from what was assumed in the yields used here (see Table 2 and 3). Three main degradation paths exist for CF3CHO: reaction with OH, photolysis, and hydrolysis. TFA yields from the OH reaction are small, whereas hydrolysis leads mostly to TFA and no TFA is formed in the photolysis path. However, the latter gained recent attention due to the potential of forming HFC-23 (a very stable compound with a large GWP). Hence, lifetimes of CF3CHO lifetimes were recently re-evaluated (Baumann et al., 2025; Sulbaek Andersen et al., 2023; Nielsen et al., 2025). The updated results suggest that previously used TFA yields may be too small and that yields will strongly vary across the troposphere. Even with larger yields the degradation of HFCs along this path may not add considerable amounts to TFA deposition (compare Table 2). However, increased TFA yields from the degradation of HFOs (other than 1234yf) could likely explain gaps between observed and simulated TFA deposition rates.*

More generally, tables and figures could be better placed throughout the manuscript (i.e. near the text the references them) and more accessible for those with colour-blindness (though I appreciate this is difficult to achieve with figures such as those presented here).

We admit that the placement of figures and tables was not optimal in the submitted manuscript. However, this should be dealt with at the time of typesetting. All figures have been re-visited in terms of readability and accessibility for color-blindness and should now meet the suggestions by Copernicus.

Specific comments:

Line 7: unnecessary '-'

Removed.

Line 8: Where does the 60-70% come from? From the other % range seems like it should be 52-71%?

We admit that the given ranges were not explained well due to the brevity of the abstract. The values give the range across the precipitation sites on the Swiss Plateau. The lowest (highest) ranges of the individual contributions don't have to add up to the overall ranges, since the sites with the lowest contribution from HFOs does not necessarily have the lowest contribution from long-lived compounds. However, we rephrased the sentence, clarifying what the ranges represent and giving the mean across the sites in addition.

*"Simulated atmospheric degradation of known TFA precursors accounted for 63 (58-70) % of the observed deposition (48 (41-54) % hydrofluoroolefins and 15 (12-18) % long-lived fluorinated gases; mean and range across sites) for sites on the Swiss Plateau."*

Line 11: Presumably this value refers to total atmospheric deposition (wet + dry?)

Inputs from PPP and veterinary pharmaceuticals do not enter through the atmosphere but through degradation in the soils. We rephrased to make this more obvious. For the case that the comment referred to L9, we also added '(wet and dry)' in front of deposition.

*"In Switzerland, atmospheric (wet+dry) deposition amounted to 24.5+/- 9.6 Mg yr$^{-1}$, whereas TFA terrestrial inputs from the degradation of plant protection products (PPP) and veterinary pharmaceuticals in soils, estimated from the literature, ranged from 3.9 to 13.2 Mg yr$^{-1}$, depending on the assumption on degradation efficiency. TFA inputs from the degradation of PPP dominated 2-3 times over atmospheric deposition in Swiss croplands."*

Line 14: 1990s instead of 1990ies (also occurs …)

Replaced everywhere.

Line 24: There are some reports that suggest TFA may bioaccumulate in plant material

This was expressed in the following sentence that referred to the accumulation in leaf material and plant extracts, most likely due to water transpiration, while icons (trifluoroacetate) remain in the plant. We rephrased somewhat to improve the link between both sentences.

" *Although TFA does not generally bioaccumulate, elevated TFA concentrations were, however, observed in leaf material and plant extracts (references)* ".

Line 39: 'As a consequence' as opposed to 'In consequence', 'potential risk' as opposed to 'risk'

We would like to thank the referee for these suggestions and adopted them.

Line 56: () within ()

This was fixed in the revised manuscript.

Line 117: Is the conducted test sufficient to prove the veracity of the archived samples?

See major comment above.

Line 169: Error of 10% for the emissions estimates – please justify this number?

The 10 % uncertainty estimate for the average hemispheric deposition rate is fully based on the uncertainty estimate of the global emissions of long-lived compounds. These uncertainties were estimated as part of the global inverse modelling calculations with the 12-box model and incorporate uncertainty due to atmospheric observations, the transport model and OH lifetime (see Henne et al., 2025 and Rigby et al., 2008). We slightly rephrased the paragraph for clarification.

"*The uncertainty on the estimated average hemispheric deposition rate will mainly be driven by the uncertainty of the emissions. The latter was estimated as part of the 12-box model simulations and, as an a posteriori estimate of a Bayesian inversion, includes contributions from the observational uncertainty and transport model uncertainty. Furthermore, uncertainties of the OH lifetime are considered. For details on the 12-box model please refer to Henne et al. (2025) and Rigby et al. (2008).*"

Line 197: You should justify why the model was run for 60 days, as from a trajectory viewpoint it seems far too long to give a reliable distribution

Please note that trajectories in FLEXPART should be seen from a stochastical point of view. Although, an individual trajectory cannot be expected to give a very good representation of an air mass transport after 60 days, FLEXPART represents air masses by a multitude of model particles which undergo both, transport by the mean wind as well as turbulent and convective transport simulated as a stochastic process. Here, we represent HFO-1234yf emissions from Europe within an individual month with 10 million released model particles. On average, this corresponds to approximately 5 model particles released per day from grid cells that contain any HFO emissions (at a spatial resolution of 0.1° x 0.1°). However, the number of released particles is kept proportional to the mass emissions in each grid cell, meaning that considerably more model particles are released in areas with high emissions. We agree that the number of model particles may is still too small to interpret simulated concentrations and deposition rates on a sub-daily scale, but the good agreement of simulated and observed HFO-1234yf at daily resolution (Figures 5 and A4), indicates that the approach is robust on this time scale. We modified some of the model description in order to emphasize the stochastical approach.

*"As a Lagrangian particle dispersion model, FLEXPART does not forecast concentrations in a spatially-fixed grid, but along a multitude of trajectories of the flow field assuming a Markov process to describe turbulent transport. Trajectories were calculated for air parcels released at the model's surface and representing a given amount of emitted HFO-1234yf (see below). Individual air parcels were tracked for 60 days in the atmosphere undergoing mean transport and turbulent and convective dispersion. The chemical composition of these air parcels was simulated according to the above description. Gridded concentrations and deposition rates were then obtained by sampling all air parcels within a given grid cell or column and during a respective averaging interval."*

Line 241: What is the timescale of hydrolysis of TFF to TFA?

In the model this happens instantaneously with a model time step of 600 seconds as soon as cloud water is present. This was already mentioned on line 187, but we now added the values of time step to clarify. This model time step is much longer than the time for complete hydrolysis of TFF with a hydrolysis rate of 150 $s^{-1}$ (George et al., 1994). The revised text reads:

*"The hydrolysis of TFF to TFA was considered to be complete within a single model time step of 600 seconds (compared to a hydrolysis rate of 150 $s^{-1}$ George et al., 1994), too short for the current model scale to allow for quantitative sensitivity tests and to assess it as a source of uncertainty."*

Line 247: Why did you use The Henry's Law constant of $HNO_3$ when there are reported values for TFA? There is uncertainty in the TFA value that may mean it diverges from the $HNO_3$ value. Given the importance of this parameter in wet deposition and the importance of this loss to the atmospheric fate of TFA, the uncertainty of this would be a useful thing to explore. Regardless, the value used should be given in the text with ref

We agree that the uncertainty associated with the wet deposition grants further exploration. We already mention that with the revised deposition scheme in the latest FLEXPART release (version 11, Bakels et al., 2024) additional studies should be carried out. In addition, and given the large Henry's Law constants involved, meaning that most TFA will go into the aqueous phase, it is probably not these constants themselves that introduce uncertainty in wet deposition, but rather simulated cloud water content and precipitation rates (the latter determining below cloud scavenging in the current parameterization in FLEXPART as already discussed in the manuscript). We used Henry's Law constant for $HNO_3$, since this was done in several previous studies. The employed value of the effective Henry's Law constant of $1 \times 10^{14}$ M atm$^{-1}$ goes back to the value published for $HNO_3$ in the original publication of FLEXPART's wet deposition scheme (Wesely, 1989), which was calculated with near-neutral pH. The effective Henry's Law constant for TFA at near-neutral pH is considerably smaller ($2.7 \times 10^{10}$ M atm$^{-1}$, based on H=8900 M atm$^{-1}$ and pKa=0.52 for TFA; Burkholder et al., 2015). Currently, FLEXPART does only employ a Henry's Law constant that is constant and does not vary with temperature or pH of cloud droplets. We added this information in the manuscript and emphasized this source of uncertainty and its improved treatment in future studies.

*"Both processes are represented in FLEXPART, assuming that TFA behaves like nitric acid (employing a constant effective Henry's Law constant of $1 \cdot 10^{14}$ M atm$^{-1}$, Wesely, 1989; Stohl et al., 2005), which corresponds to the vast majority of TFA residing in the liquid phase and, hence, fast washout through in-cloud wet scavenging. FLEXPART treats in-cloud and below-cloud scavenging of gases separately, with the latter only depending on precipitation rates."*

*"The latest version of FLEXPART (v11, Bakels et al., 2024) features an improved description of atmospheric deposition and future simulations should benefit from this update and should also consider a more specific and temperature dependent Henry's law constant for TFA."*

Line 255-275: There should be mention of recent publications relating to $CF_3CHO$ that dispute TFA yields, specifically the 4-60% value for HFO-1336mzzZ, and introduce extra uncertainty that hasn't been accounted for in this study as it stands

*This relates back to the major comment above. We now reflect on the role of CF3CHO degradation pathways in the discussion. We also emphasize that a re-evaluation will be especially important for the HFOs (other than HFO-1234yf) as their contribution may thus be similar to that of HFO-1234yf.*

Line 274: Typo 'emissions'

Fixed.

Line 280: Are the samples below the estimated quantification limit considered in the average? How can you quantify a value below the limit of quantification? What is the limit of detection? How is the limit of quantification derived?

Few samples below LOQ are present in the presented TFA dataset (97 of 1570 samples). These were estimated as 0.04 ng L$^{-1}$ and were considered as such in any aggregation/averaging. The LOQ was estimated in Scheurer et al. (2017) following DIN 32645 (DIN, 2008) by equidistant calibration. No level of detection was defined in their study. We slightly revised our statement pointing to the original publication of the analytical method.

*"All water samples were filled in PE centrifuge vials prepared at FOEN, RüS or Eawag and shipped to the DVGW-Technologiezentrum Wasser (German Water Centre) in Karlsruhe, Germany, where they were analysed by ion exchange liquid chromatography (LC) coupled to electrospray tandem mass spectrometry (MS-MS) with a limit of quantification (LOQ) determined as 50 ng L$^{-1}$ (Scheurer et al., 2017). Any values reported below the LOQ (97 of a total of 1570 samples) were given as 0.04 ng L$^{-1}$ and were treated as such in any averaging"*

Line 281: Is this 0.58 ± 0.58 correct?

Yes.

Line 283-303: Are you using T-dependent rate constants for HFO+OH? Some of the HFOs you are considering have a negative relationship between k and T

Yes. The temperature dependency was considered for the explicitly treated HFO-1234yf. For the smaller contributors (HFO-1234ze(E), HFO-1336mee(Z), and HCFO-1233zd(E)) no explicit simulations were done and, hence, their potentially opposite lifetime dependency on temperature was not considered. To clarify, we added the utilized rate constants for HFO-1234yf to the text:

"*HFO-1234yf degradation due to reaction with OH and Cl is described following the scheme and rate constants ($k_{OH}$ =1.26·10$^{-12}$exp(−35/T) cm$^3$molec$^{-1}$s$^{-1}$ and $k_{Cl}$ = 7.03·10$^{-11}$ cm$^3$molec$^{-1}$s$^{-1}$) suggested by Hurley et al. (2008).* "

Line 311: Could deposition not also be a driver of load seasonality (in addition to discharge seasonality)?

Yes, this is what we wanted to express with these two sentences. Although, we don't directly see the seasonality in rainwater concentrations reflected in the river concentrations, we see the same seasonality in the loads. Hence, additional run-off (meltwater and groundwater) dilute the precipitation concentration signal. We slightly rephrased to clarify our intention:

"*However, seasonality of TFA loads (concentration multiplied by flow rate) with a summertime peak was observed at S-Chanf and Diepoldsau, where there are no greater lakes in the catchment upstream and we can expect little delay between the concentration maximum in precipitation and that in river discharge (Fig. 3b). The latter seasonality of loads is clearly related to the discharge seasonality at these sites, with greater discharge in the summer, partly from snow and glacier melt, offsetting the concentration signal.*"

Line 327-329: 'underlines' and 'underlining' should be replaced with 'supports' and 'reinforcing'

We would like to thank the referee for these suggestions, which we adopted accordingly.

Line 330-334: Does this evidence not serve as an argument against pre known-precursor TFA?

The glacial ice dates from 1892 to 1978. The absence of TFA (i.e., concentrations below LOQ) in these samples underlines that there were no major atmospheric precursors contributing to TFA deposition before 1978. However, about a decade later TFA can be quantified at least in summer-time precipitation samples. The precipitation weighted mean TFA from the archived precipitation samples collected in 1986/87 was not much higher than the LOQ, 0.08 ug L-1 (N=25), for which nine samples were below the LOQ (see section 3.4). Hence, we can conclude that the unknown pre-cursor gained importance in the 1980ies but at a relatively slow rate. We added this information in the discussion of the unknown precursor in section 3.5.1.

*Furthermore, and as described above, glacial ice dating back to 1978 did not contain TFA above the LOQ, which would imply an earliest possible onset and slow growth of a significant contribution of an atmospheric precursor to TFA deposition in the Swiss Alps at the beginning of the 1980s.*

Line 343: 'No significant TFA gradients with depths were detected in the other lakes' so concentrations at depth were the same as the surface? Were the values elevated?

Samples were taken both at surface and lake bottom, as indicated in Table 1. Both for Lake Zurich and Lake Geneva no significant concentration difference were observed. We rephrased to clarify our statement:

" *No significant TFA differences between lake surface and lake bottom were observed in the Lake Zurich and Lake Geneva (Table 1).*"

Line 383: Replace 'the latter' with 'this'

Done.

Line 391: 'large HFO emissions in the Po Valley' ? Are these emissions simulated in the model at an appropriate resolution to be able to make this claim?

Emissions are provided to the model at a resolution of 0.1° x 0.1° (see section 2.3.2 and Figure A2). At this resolution the emission maximum in the Po Valley is well resolved. No changes to the text seemed required.

Line 400: 'compared with' instead of 'compared to'

Done.

Line 409: Remove 'as well' and replace with also earlier in the sentence

Done.

Line 415: It also suggests that if these unknown sources exist, they are likely to be anthropogenic

We agree with this conclusion and have added it to the sentence.

*"The explained fraction was considerably smaller for two sites in more urban/suburban environments (Bern, Basel-Binnigen), which may indicate additional anthropogenic primary sources of TFA to the atmosphere."*

Line 426: What is the estimated combined uncertainty of the simulated TFA deposition? It should be reported here along with the specific contributions of the TFA yield vs the precursor emissions

The estimated combined uncertainty of the simulated TFA deposition was given in Figure A5. This was not very clear and has now been added to the sentence and an average uncertainty is now given for the Swiss Plateau sites as well.

*The estimated combined uncertainty of the simulated TFA deposition, accounting for uncertainty in TFA yields and precursor emissions, was about 66 % for the precipitation sites on the Swiss Plateau (compare Fig. A5) and cannot fully explain the unaccounted fraction of TFA in the observations.*

Line 430-431: This sentence 'faster (slower) ... increased (decrease)' is not worded clearly.

We rephrased to:

*"A 10 % faster OH degradation resulted in an increase of TFA deposition in the Swiss domain of 4 %. Vice-versa, a 10 % slower OH degradation decreased deposition by 4 %."*

Line 448: 'which is in the range smaller 0.3%' does not make sense and there is also an unclosed '('

The threshold of 0.3 % was given as an upper limit for individual compounds, whereas the example for the most important HFC yields a smaller contribution. We rephrased as follows:

*"However, since hemispheric background mole fractions determine the TFA deposition, the domestic fraction should be similar to the Swiss contribution to global emissions of these compounds, which is smaller 0.3 % for*

*the long-lived compounds considered. In the case of HFC-134a, Swiss emission of 410 Mg yr$^{-1}$ (Swiss Federal Office for the Environment, 2024b) in 2022 can be contrasted by global emissions of 250 Gg yr$^{-1}$ (Henne et al., 2025), hence, a Swiss contribution of 0.16 %.*"

Line 451: 'For non-atmospheric sources of TFA'

Done.

Line 484: Need an 'is' between 'discharge' and 'below'

Done.

Line 489: This should be more specific e.g. 'PPPs may be a more relevant source of TFA to environmental aqueous phases (i.e rivers and lakes) in Switzerland'

We would like to thank the referee for this suggestion and rephrased it accordingly.

"*These results indicate that over Swiss cropland PPPs may be the more relevant source of TFA to environmental aqueous phases (i.e., rivers, lakes and groundwater).*"

Line 494: Remove () from reported flux to be consistent

Done.

Line 503: Missing '('

Done.

Line 511: TFA in rainwater in Bayreuth?

Yes, in rainwater. The information was added to the text.

Line 514-518: Clarify that you're talking about TFA concentration in rainwater and in archived plant material

This paragraph reviewed earlier TFA in rainwater concentrations reported in the literature and compared them to the archived rainwater samples as analyzed for this study. There was no mention of archived plant material. Hence, no changes were made to the text.

Line 528: Was 2019 when HFO use started to increase exponentially? If yes, I would explicitly make this link

Yes, HFO emissions increased strongly after 2019 as indicated in Figure A1. The growth was more linear than exponential, but since we did not use the latter term for the increase in TFA in precipitation either, the link between TFA in precipitation and HFO emissions remains fairly obvious and was added to the sentence.

"*After 2019, the increase in TFA concentrations in precipitation accelerated reaching mean (all sites) concentrations of 0.643 µg L$^{-1}$ in 2023 (PWM: 0.496 µg L$^{-1}$), which, qualitatively, is in line with a strong increase of HFO emissions in the same period (see Fig. A1).*"

Line 542: You should be specific as to what other sources you are suggesting (agricultural run-off? Natural sources?)

The remainder of the paragraph explains our suggestion that these emissions are from aluminum smelting. We added 'industrial' in this sentence to clarify the link with the following.

Line 548: Do these processes release fluorocarbons into the environment via wastewater etc only or is there a potential gaseous emission?

Both pathways are possible and the widespread fluorocarbon pollution on the grounds of the former aluminum smelter suggests considerable transport through the atmosphere. However, no quantitative knowledge of the importance of the pathways exists. The text was modified to acknowledge both possibilities.

Line 627: Also there's massive uncertainty in the yields?

We feel that the possibility of uncertainty in the yields was already addressed in the previous sentence and does not require repetition in this sentence, which points at the temporal and spatial variability of yields. No changes were made to the text.

Figure 2 & 3: Additional x-ticks representing the months or every 3 months would make these figures easier to interpret where variations are not exactly halfway through the year

Figures 2 and 3 were completely revised also to make them more suitable for color-blind readers. Instead of 3-monthly minor ticks, 6-monthly minor ticks were added and since the individual sites were more separated in the display, it should be much easier to follow the annual cycle.

Table 2 & 3: Should present the 3 yields used rather than the range, lifetime of '1' should be reported as '1.0' for consistency, last column in each needs () around units

The yields are now presented as mean (min – max). Values of 1 are now given with decimal point and all units are given in braces.

**References**

Bakels, L., Tatsii, D., Tipka, A., Thompson, R., Dütsch, M., Blaschek, M., Seibert, P., Baier, K., Bucci, S., Cassiani, M., Eckhardt, S., Zwaaftink, C. G., Henne, S., Kaufmann, P., Lechner, V., Maurer, C., Mulder, M. D., Pisso, I., Plach, A., Subramanian, R., Vojta, M., and Stohl, A.: FLEXPART version 11: improved accuracy, efficiency, and flexibility, Geosci Model Dev, 17, 7595-7627, doi: 10.5194/gmd-17-7595-2024, 2024.

Burkholder, J. B., Cox, R. A., and Ravishankara, A. R.: Atmospheric Degradation of Ozone Depleting Substances, Their Substitutes, and Related Species, Chemical Reviews, 115, 3704-3759, doi: 10.1021/cr5006759, 2015.

DIN, 2008. Chemical analysis - decision limit, detection limit and determination limit under repeatability conditions - terms, methods, evaluation. DIN 32645, 2008 - 2011.

George, C., Saison, J. Y., Ponche, J. L., and Mirabel, P.: Kinetics of Mass-Transfer of Carbonyl Fluoride, Trifluoroacetyl Fluoride, and Trifluoroacetyl Chloride at the Air/Water Interface, Journal of Physical Chemistry, 98, 10857-10862, doi: 10.1021/j100093a029, 1994.

Henne, S., Vollmer, M. K., Guillevic, M., Schlauri, P., and Reimann, S.: The Annual Kigali-Index (AKI): a climate change indicator in support of the hydrofluorocarbon phase-down of the Montreal Protocol, in preparation, 2025.

Rigby, M., Prinn, R. G., Fraser, P. J., Simmonds, P. G., Langenfelds, R. L., Huang, J., Cunnold, D. M., Steele, L. P., Krummel, P. B., Weiss, R. F., O'Doherty, S., Salameh, P. K., Wang, H. J., Harth, C. M., Mühle, J., and Porter, L. W.: Renewed growth of atmospheric methane, Geophys. Res. Lett., 35, L22805, doi: 10.1029/2008GL036037, 2008.

Wesely, M. L.: Parameterization of Surface Resistances to Gaseous Dry Deposition in Regional-Scale Numerical-Models, Atmos. Environ., 23, 1293-1304, doi: 10.1016/0004-6981(89)90153-4, 1989.

---

## Author Comment (AC3)

**Reply to 'Comment on egusphere-2025-2861' by Tim Wallington**

Comments are copied in plain text.

Replies are given in blue.

*Changes to manuscript are highlighted in green.*

Henne et al. report the results of a comprehensive set of measurements of trifluoroacetate (TFA) in surface waters in Switzerland. There is substantial research interest in the sources, concentrations, and effects of trifluoroacetate in the environment. This paper is an important contribution to better understanding the levels, sources, and trends of trifluoroacetate in the environment. We have the following comments for the authors to consider.

We would like to thank the authors for their insightful comments and have adopted most of the suggestions as outlined below.

First, the United Nations Environmental Effects Assessment Panel of which some of us are members has assessed that the risks to human and ecosystem health from trifluoroacetate formed as a degradation product of ODS replacements are currently *de minimis* (Neale et al., 2025). For balance in the introduction where the toxicological effects are discussed the authors may wish to mention the assessment of the UNEP panel and note the need for further work to reconcile the divergent assessments in the literature.

We added a reference to the report/publication of the UNEP Assessment panel and repeat their main conclusion in the introduction.

*"The UNEP Environmental Effects Assessment Panel reviewed the risks of TFA to ecosystems and human health and conclude that at current TFA concentration levels the risk to humans is* de minimis *(Neale2025 et al., 2025)".*

Second, there are new measurements published by the German UBA (Umweltbundesamt) of trifluoroacetate in samples from the Atlantic Ocean collected from surface waters (n = 33) and from seven distinct depth profiles (n = 41) in 2022–23. Using these data, Neale et al. estimated that the mass of TFA measured in the oceans in the late 1990s and early 2000s, assuming even distribution of 200 ng $L^{-1}$, was about 500–1000 times higher than the estimated total anthropogenic TFA input to the environment (including Montreal Protocol gases, pesticides, pharmaceuticals, and industrial uses) over the period 1930–1999. The evidence for the contribution of one or more natural source(s) of TFA to the marine environment is relevant and should be mentioned.

We would like to thank the reviewers for their comment and acknowledge that we did not mention the UBA report so far because it is only available in German (besides the summary). We also don't think that the few samples for the deeper ocean are sufficient to estimate global mean ocean concentrations of 200 ng $L^{-1}$. More and representative observations in all major ocean basins would be necessary to arrive at this conclusion. Moreover, samples presented by Frank et al. (1996) were taken in 1995 near to the coast. Therefore, the latter TFA ocean concentrations rather overestimate the deep sea/open ocean TFA concentration, as wastewater (e.g. Nödler et al., 2014) and other inputs (e.g. groundwater exfiltration with impact from agriculture) may increase the coastal concentration of micropollutants in seawater. In the end, there is no consensus on the possibility of a natural TFA source in the oceans and we had acknowledged this by citing the study by

Lindley (2023), on the one hand, and the arguments of Joudan et al. (2021) on the other hand. Nevertheless, we added the results from the UBA report to the introduction, but without the conclusion that mean TFA concentrations in the ocean are at 200 ng L$^{-1}$, which the UBA authors also did not suggest but rather speculated on mechanisms that would bring TFA down towards the deeper ocean on shorter time scales (*Atlantic Meridional Overturning Circulation, Dense Shelf Water Cascading, sedimentation of particles and dead organisms*).

*"Earlier studies have speculated about the existence of natural sources of TFA (i.e., deep-sea vents Scott et al., 2005). More recently (2022/2023), TFA concentrations of ocean samples down to a depth of 4590 m in the North Atlantic were estimated in the range of 237 to 294 ng L$^{-1}$, with slight decreases with depth and higher TFA concentrations at the ocean surface, 260 – 306 ng L$^{-1}$ (UBA, 2024). A comparison with earlier observations is hindered by questions of measurement quality and validity."*

Third, using the flux of trifluoroacetate (TFA) measured at several locations on the Rhine River in the Netherlands (2017-2023), an average flux into the Rhine basin of ca 0.5 kg TFA km$^{-2}$ yr$^{-1}$ can be estimated (Neale et al., 2025).  This is of a similar magnitude to that estimated in Switzerland by Henne et al.  There was no discernable trend in the flux of TFA in 2017-2023 suggesting either that degradation of HFO-1234yf was not a major contributor to TFA in the Rhine basin for 2017–2023, or that its increasing contribution was masked by compensating decreases in contributions from other sources. A discussion of how their findings compare with those of Neale et al. (2025) would be a useful addition.

We would like to thank the authors for pointing us to the TFA measurements at the lower Rhine in the Netherlands as published in the RIWA annual reports (Dutch only). However, we feel that the assessment of these concentration measurements in Neale et al. (2025) over-simplifies the complexities of the TFA budget in the large Rhine basin as it does not consider other hydrological pathways other than river run-off (e.g., groundwater formation, water extraction for irrigation) and lateral TFA fluxes. Scheurer et al. (2017) had analyzed TFA fluxes in the Rhine catchment and identified massive industrial releases in the river Neckar draining into the Rhine, but also concluded that there were additional industrial or waste water sources required to maintain and even increase TFA concentrations further downstream. Very similar to the TFA observations in the Netherlands, they reported TFA concentrations of 1.3 ug L$^{-1}$ at the Rhine in Düsseldorf (close to the Dutch boarder). Following the publication by Scheurer et al. some attention was given to abate the TFA sources in Germany and it is very likely that the absence of a trend in TFA concentrations in the Rhine at Lobith (Netherlands) is due to a combination of decreasing industrial releases and increasing atmospheric inputs. However, without detailed hydrological modelling for the Rhine basin it seems impossible to better quantify these inputs. We re-evaluated the data mentioned by Neale et al. (2025) for the Rhine River water measurements at Lobith. For the years 2021-2023 (as in our study) we estimate an average TFA load of 79 Mg yr$^{-1}$, based on monthly TFA concentrations and water flow data. This compares to 93 Mg yr$^{-1}$ in Neale et al., who applied a single average TFA and flow to derive the load. Please note that month-to-month variability in the loads is large and depends considerably on water levels. This variability also complicates the robust estimation of a trend. If we make the same assumption as in Neale et al. that the TFA load of the Rhine can be equated to the total TFA input through atmospheric deposition in the Rhine catchment (185'000 km$^2$), we arrive at an average deposition flux of 0.43 kg km$^{-2}$ yr$^{-1}$, 0.5 kg km$^{-2}$ yr$^{-1}$ in Neale et al. This compares to a simulated average deposition flux of 0.46 kg km$^{-2}$ yr$^{-1}$ in the Rhine catchment directly taken from our model runs. Hence, the conclusion that atmospheric inputs dominate the TFA budget of the Rhine might be drawn. However, the average simulated TFA rainwater concentration for the same area was only 0.43 ug L$^{-1}$, compared to 1.2 ug L$^{-1}$ in the Rhine

at Lobith. This concentration enhancement still points towards additional sources and the need for more detailed hydrological modelling, which is beyond the scope of this study. Neale et al. (2025) used the discussion of the TFA concentrations at Lobith to dispute rising atmospheric TFA inputs. Given the results of our current analysis, which use a much more direct way to quantify atmospheric TFA inputs (i.e., archived precipitation samples), there can be no doubt about increased inputs after the market introduction of HFOs (compare Fig. 10, which uses a logarithmic scale). Furthermore, Freeling et al., (2020) reported observed TFA wet deposition rates of around 0.2 kg km$^{-2}$ yr$^{-1}$ for 3 sites in the Rhine catchment (see ES, SU, WK in their Table 2) and for the year 2018. Compared to our recent observations and simulations, this is in line with a strong increase in deposition in recent years. Hence, we don't think the discussion of the larger Rhine catchment and the connected uncertainties mentioned above, would add to our current manuscript. Nevertheless, we added the observation of Scheurer et al. (2017) for the Rhine in Basel to Fig. 11 and added the following general discussion of TFA in the Rhine.

" *Since 2000 concentrations rose exponentially (Fig. 11, note logarithmic y-axis). In 2017, Scheurer et al. (2017) reported TFA concentrations in the Rhine in Basel of 0.4 µg L$^{-1}$, in line with the long-term increase and current (2021-2023) levels of 0.65 µg L$^{-1}$ (Tab. 1). Further downstream (Scheurer et al., 2017) observed strongly enhanced TFA concentrations in the Rhine (up to 1.3 µg L$^{-1}$) and traced these to industrial sources and WWTP discharges. In the Netherlands, TFA is monitored by RIWA at several locations along the Rhine. Annually reported concentrations have largely remained above 1 µg L$^{-1}$ since 2017, although peak concentrations (as expected from industrial discharges) have decreased in recent years (Fig. 1.18 in RIWA-Rijn, 2024).*"

Fourth, it should be possible to propagate the error bars for the relevant parameters for both the precursor measurements and the TFA measurements/modelling results. This would allow for better comparison of the contributions to TFA accounted for and the "unaccounted" remainder. On the same note, greater clarity on how the TFA deposition fluxes from the individual precursors were calculated would be beneficial for the reader, for example which molar yield of TFA from HFO-1233ze was used for the calculations. This combined information would be informative and most interesting. Possibly, it could hint at additional atmospheric sources of TFA or lacking/incorrect understanding of the atmospheric oxidation chemistry involved.

We did propagate the uncertainties of the components of our calculations of TFA deposition at the individual observations sites wherever this was possible. This result was given as Figure A5 and briefly discussed in section 3.2. However, no complete description of the uncertainty calculation was given, and we now provide this information along with the Figure in the appendix. Other sources of uncertainty that cannot be quantified as easily (all model related elements like transport, reaction and deposition rates, representativeness of simulations are discussed qualitatively in the text. The following text on the uncertainty calculation was added to the appendix:

"*The uncertainty of simulated TFA deposition rates for both long-lived and short-lived compounds was assessed as follows. For long-lived compounds we consider two sources of quantifiable uncertainty: simulated loss rates and TFA yields. The uncertainty of the former is taken from the a posteriori uncertainty estimate of global emissions for individual compound as estimated with the 12-box model. This estimate contains the uncertainty of the atmospheric observations propagated through the inverse modelling step and an additional lifetime uncertainty (see Rigby et al., 2008). For most compounds the by far larger uncertainty originates from the TFA yields. We assume that the values given in Table 2 represent the 95 % confidence range of the yields. We use Gaussian error propagation to combine both sources of uncertainty and sum over all long-lived compounds, where we assume fully uncorrelated uncertainty between yields and emissions and full correlated uncertainties for the yields*

*between compounds, since many of these come from uncertainties of yields from intermediate compounds. Similarly, we combine the uncertainty estimate on European emissions and yields for the additional HFO/HCFOs (Table3). Not formally quantified in this calculation are other sources of model uncertainty (transport, deposition rates) and representativeness, which are discussed qualitatively in the main text. The resulting uncertainty range for the relative contributions to observed deposition rates (Figure A5) represents the 95 % confidence range of this assessment.*"

Applied TFA yields for all HFCs and the direct impact on deposition rates are summarized in Table 2. For HFOs Table 3 contains the yields (range) and the European emission estimate. The last paragraph of section 2.3.2 discusses these numbers and where they were taken from. We feel that Table 2 and Table 3 already provide the information asked for (e.g., a yield of 2 – 30% for HCFO-1233zd(E).

Fifth, the statement in the conclusions "*Therefore, it is fundamental to continue efforts to abandon the use of fluorinated compounds, wherever possible, to avoid further, continued accumulation of TFA. Both industry and policy makers are called to increase their level of ambition*" is very generalized and simplistic. It fails to consider that not all fluorinated compounds degrade to produce TFA and that the stoichiometry is such that yields are not always molar. It also fails to recognize that the $CF_3-$ group acts as a pseudo halogen that increases the efficacy of pharmaceuticals and pesticides to the direct and indirect benefits of humans and the environment. As discussed in Neale et al. (2025), TFA in the environment is present as salts that are highly water soluble and easily excreted. TFA-salts do not biomagnify in food webs and there are no known biochemicals or receptors that interact with TFA, although it is a moderately strong acid (pKa = 0.23), it is unreactive. There are wide margins of safety between current and predicted future concentrations in surface- and ground-waters levels of concern for human and environmental health. While continuous monitoring would be useful in quantifying future rates of change in concentrations, this should be focused on key matrices and should include measurements of systemic doses in the general population, such as those conducted in the NHANES program [1].

We agree with the reviewer insofar, that the reference to "fluorinated compounds" – while not irrelevant in terms of their effects on human health and the environment – is broader than the scope of our paper. We therefore adjusted the wording and refer now to "TFA and its precursors" instead. We do acknowledge the potential benefits of such compounds, why we formulated in our initial submission "wherever possible". We now provide further clarity on that point, referring to "abandoning all non-essential uses", instead.

We are aware that TFA does neither bioaccumulate nor biomagnify in food webs. However, as a persistent and mobile substance and with continuous emissions increasing environmental concentrations, it does accumulate in organisms. Together with its currently established classification as aquatic chronic 3 and its postulated classification as toxic to reproduction (Category 1B) as well as very persistent and very mobile (vPvM), a thorough application of the precautionary principle is warranted. Also the margins of safety (which under the current classification and existent guideline values in drinking water are on the order of 10) are decreasing and will continue doing so, until the effect of future regulation becomes visible.

We acknowledge the recommendation to place a focus on measurements in the general population and insert the corresponding remark.

*Therefore, it is fundamental to continue efforts to abandon all non-essential uses of TFA and its precursors, to avoid further, continued accumulation of TFA. Both industry and policy makers are called to increase their level of ambition. Also, continued continent-wide monitoring, including human biomonitoring, will be necessary to*

*surveil progress and to further improve both our understanding of TFA budgets and our ability to forecast future burdens by atmospheric simulations.*

**References**

Frank, H., Klein, A., and Renschen, D.: Environmental trifluoroacetate, Nature, 382, 34-34, doi: 10.1038/382034a0, 1996.

Neale et al. Environmental consequences of interacting effects of changes in stratospheric ozone, ultraviolet radiation and climate: UNEP Environmental Effects Assessment Panel, Update 2024, Photochem. Photobiol. Sci., 24, 357 (2025). https://doi.org/10.1007/s43630-025-00687-x

Nödler, K., Voutsa, D., and Licha, T.: Polar organic micropollutants in the coastal environment of different marine systems, Marine Pollution Bulletin, 85, 50-59, doi: 10.1016/j.marpolbul.2014.06.024, 2014.

UBA. (2024). *Untersuchung von aktuellen Meerwasserproben auf Trifluoressigsäure. TZW: DVGW-Technologiezentrum Wasser, Karlsruhe, Umweltbundesamt, Dessau-Roßlau, Germany.* https://www.umweltbundesamt.de/publikationen/untersuchung-von-aktuellen-meerwasserproben-auf